

# Error curves for evaluating the quality of feature rankings

Ivica Slavkov[1], Matej Petković[1,2], Pierre Geurts[3], Dragi Kocev[1,2] and Sašo Džeroski[1,2]

[1] Jozef Stefan Institute, Ljubljana, Slovenia
[2] Jozef Stefan International Postgraduate School, Ljubljana, Slovenia
[3] Université de Liège, Liège, Belgium

## ABSTRACT

In this article, we propose a method for evaluating feature ranking algorithms. A feature ranking algorithm estimates the importance of descriptive features when predicting the target variable, and the proposed method evaluates the correctness of these importance values by computing the error measures of two chains of predictive models. The models in the first chain are built on nested sets of top-ranked features, while the models in the other chain are built on nested sets of bottom ranked features. We investigate which predictive models are appropriate for building these chains, showing empirically that the proposed method gives meaningful results and can detect differences in feature ranking quality. This is first demonstrated on synthetic data, and then on several real-world classification benchmark problems.

## INTRODUCTION

In the era of data abundance, we face high-dimensional problems increasingly often. Sometimes, prior to applying predictive modeling (e.g., classification) algorithms to such problems, dimensionality reduction may be necessary for a number of reasons, including computational reasons. By keeping only a limited number of descriptors (features), a classifier can also achieve better predictive performance, since typically, a portion of the features strongly influence the target variable, and the others can be understood as (mostly) noise. This dimensionality reduction corresponds to the task of feature selection (*Guyon et al., 2002*). A task related to it is feature ranking. This is a generalization of feature selection where, in addition to simply telling apart relevant features from irrelevant ones (*Nilsson et al., 2007*), one also assesses how relevant are they for predicting the target variable.

In machine learning, feature ranking is typically seen either as a preprocessing or as a postprocessing step. In the former case, one actually tackles the feature selection problem by first computing the feature relevance values, and then keeping only the features whose relevance is above some user defined threshold. In the second case, feature ranking is obtained after building a predictive model in order to explain it, for example (*Arceo-Vilas et al., 2020*). For black box models, such as neural networks, this may be the only way to understand their predictions.

Corresponding author
Matej Petković, matej.petkovic@ijs.si

In some application domains, such as biology or medicine, feature ranking may be the main point of interest. If we are given data about the expression of numerous genes, for a group of patients, and the patients' clinical state (diseased/healthy), one can find good candidate genes that influence the health status of the patients, which gives us a deeper understanding of the disease.

Due to the prominence of the feature ranking task, there exist many feature ranking methods. Simpler methods assess the relevance of each feature independently ignoring the other features ($\chi^2$ statistics, mutual information of the feature and the target variable) and their possible interactions. A prominent example that shows the myopic nature of such approaches is the case when the target variable $y$ is defined as $y = \text{XOR}(x_1, x_2)$ where $x_1$ and $x_2$ are two binary features. Ignoring $x_1$ when computing the relevance of $x_2$ (and vice-versa) would result in assessing $x_1$ as completely irrelevant, that is, as random noise. More sophisticated methods assess relevance of each feature in the context of the others. They are typically based on some predictive model, for example, Random Forest feature ranking (*Breiman, 2001*), or optimization problem (*Nardone, Ciaramella & Staiano, 2019*), but not necessarily, cf. for example, ReliefF (*Robnik-Šikonja & Kononenko, 2003*) and the work of *Li & Gu (2015)*.

However, there is no unified definition of feature importance, and actually, every feature ranking algorithm comes with its own (implicit) definition. Therefore, different methods typically introduce different feature importance scores: Deciding which of them is the best is a very relevant, but also very challenging task that we would like to address in this article. More precisely, we continue and extend our previous work (*Slavkov et al., 2018*), where we proposed and evaluated a quantitative score for the assessment of feature rankings. Here, we propose a new feature ranking evaluation method that can evaluate feature rankings in a relative sense (deciding which of the feature rankings is better), or in an absolute sense (assessing how good is a feature ranking). The method is based on constructing two chains of predictive models that are built from the top-ranked and bottom-ranked features. The predictive performances of the models in the chain are then shown on graphs of so called forward feature addition (FFA) and reverse feature addition (RFA) curves, which reveal how the relevant features are distributed in the ranking(s). An important property of the proposed method is that it does not need any prior ground truth knowledge of the data.

We investigate the performance of the proposed evaluation approach under a range of scenarios. To begin with, we prove the potential of the FFA and RFA curves by using them in setting which employs synthetic data. Next, we investigate the use of different types of predictive models for constructing the curves, thus considerably extending the preliminary experiments by *Slavkov et al. (2018)*. Furthermore, we apply the proposed evaluation approach to a large collection of benchmark datasets. Compared to *Slavkov et al. (2018)*, we have included 11 new high-dimensional datasets. The results of the evaluation, in a nutshell, show that the FFA and RFA curves are able to discern the best ranking among multiple proposed feature rankings.

The remainder of this article is organized as follows. "Related Work" outlines related work, "Method for Evaluating Feature Rankings" describes in detail the proposed method for constructing error curves. Next, "Empirical Evaluation of FFA/RFA Curve Properties" discusses the properties of the error curves when applied to synthetic data. We then give the results of the experimental evaluation on benchmark datasets in "Feature Ranking Comparison on Real-Worlds Datasets". "Conclusions" concludes with a summary of our contributions and an outline of possible directions for further work. In the appendices, we give additional information about generating synthetic data (Appendix A1), measuring distance between rankings (Appendix A2), and comparative evaluation of feature ranking methods (Appendix A3). In Appendix A4, detailed experimental results are given.

## RELATED WORK

The evaluation of feature rankings is a complex and unsolved problem. Typically, feature rankings are evaluated on artificially generated problems, while evaluation on real world problems remains an issue approached indirectly. To begin with, when the ground truth ranking is known, one can transform the problem of feature ranking evaluation into an evaluation of classification predictive model (*Jong et al., 2004*) as follows. First, a ranking is computed. Then, for every threshold, the numbers of relevant features (true positives) and irrelevant features (false positives) with the feature relevance above the threshold are computed. From these values, a ROC curve can be created and the area underneath it computed.

Another possible approach is to compute separability (*Robnik-Šikonja & Kononenko, 2003*), that is, the minimal difference between the feature importance of a relevant feature and the feature importance of an irrelevant feature. If this difference is positive, then the relevant features are separated from the irrelevant ones, otherwise they are mixed.

However, both approaches are more applicable to feature selection problems and are too coarse for feature rankings problem, since they only differentiate between relevant and irrelevant features. Spearman's rank correlation coefficient between the computed and the ground truth ranking might be more appropriate.

The main shortcoming of the upper approaches is that they demand the ground truth ranking. In real world scenarios, this is not known, which makes the upper approaches useless. Nevertheless, using synthetic data and the controlled environment offers a good starting point for showing the usefulness of a feature ranking evaluation method, as we shall also see later.

An approach that overcomes the issue of unknown ground truth ranking bases on selecting $k$ top-ranked features and building a predictive model that uses only these features to predict target variable. The ranking whose top-ranked features result in the model with the highest predictive performance, is proclaimed the best. Since it is now always clear which value of $k$ should be chosen, this can be done for multiple values of $k$ (*Guyon et al., 2002*; *Furlanello et al., 2003*; *Paoli et al., 2005*; *Verikas, Gelzinis & Bacauskiene, 2011*).

In addition to correctness, rankings stability is sometimes also part of the evaluation. The stability of a ranking algorithm can be measured by comparing the feature rankings obtained, for example, from the different bootstrap replicates of a dataset or from the folds in cross-validation (*Guzmán-Martnez & Alaiz-Rodrguez, 2011*; *Kalousis, Prados & Hilario, 2007*; *Jurman et al., 2008*). In *Saeys, Abeel & De Peer (2008)* both stability and predictive performance are combined into a single feature ranking quality index.

Also, notions similar to FFA curves (without any particular name, though) as the feature ranking evaluation method can be found in the literature (*Liu et al., 2003*; *Duch, Wieczorek & Biesiada, 2004*; *Biesiada et al., 2005*; *Liang, Yang & Winstanley, 2008*). However, to the best of our knowledge, there is no discussion and detailed investigation why FFA curves are an appropriate method for comparing feature rankings, nor which learning methods should (or should not) be used for constructing them.

## A METHOD FOR EVALUATING FEATURE RANKINGS

First of all, every feature ranking method should be able to tell apart relevant features from irrelevant ones (*Nilsson et al., 2007*). In addition to that, the method should order the features with respect to the target variable, awarding the most relevant ones the top ranks.

If ground truth ranking exists, the method should return this ranking in the optimal case. The worst case is more complicated and has two possible answers. One is the inverse of the ground truth ranking. However, since the ground truth ranking is typically not known in real-world scenarios, a more useful definition of the worst ranking is random ranking. This ranking also contains as little information about the distribution of the relevant features in the ranking as possible. Moreover, this distribution can be always assessed and is the cornerstone of our ranking evaluation method.

### The evaluation method

First, we define the notation used in the rest of the article: $\mathscr{D}$ denotes a dataset whose columns are input features $F_i$ that form a set $\mathscr{F}$, and the target feature $F_t$. A feature ranking method takes the dataset as an input, and returns a list $\mathbf{R} = (F_{(1)}, \ldots, F_{(n)})$ as the output, where $F_{(i)}$ is the feature with the rank $i$.

We evaluate a ranking $R$ by evaluating different subsets $S$ of features $\mathscr{F}$. This is done by building a predictive model $\mathscr{M}(\mathbf{S}, F_t)$ and assessing its predictive power. The evaluation of the predictive model provides a cumulative assessment of the information contained in the feature set $S$ and it can be quantified with an error measure $err(\mathscr{M}(\mathbf{S}, F_t))$. The question is how to generate the feature subsets from the feature ranking, so that the error estimates can provide insight into the correctness of the feature ranking and constitute an evaluation thereof.

The construction of the feature subsets should be guided by the feature ranking $\mathbf{R}$. Starting from the top ranked feature $F_{(1)}$ and going towards the bottom ranked feature $F_{(n)}$, the feature relevance should decrease. Following this basic intuition, we propose two methods for constructing feature subsets from the feature ranking: FFA and RFA.

---

**Algorithm 1** Generation of the FFA and RFA curves.

**Input:** Feature Ranking $\mathbf{R} = (F_{(1)},\ldots,F_{(n)})$, Target Feature $F_t$, type of curve (FFA or RFA)

$S \leftarrow \varnothing$

$E \leftarrow$ list of length $n$

**for** $i = 1,2,\ldots,n$ **do**

    **if** curve type is FFA **then**

        $S \leftarrow S \cup \{F_{(i)}\}$

    else

        $S \leftarrow S \cup \{F_{(n-i+1)}\}$

    $E[i] \leftarrow$ err $(\mathcal{M}(S, F_t))$

**return** $E$

---

Forward feature addition constructs the feature subsets $S^i$ by considering the $i$ highest ranked features, starting with $S^1 = \{\mathscr{F}_{(1)}\}$. The next set $S^{i+1}$ is constructed by adding the next lower-ranked feature, namely $S^{i+1} = S^i \cup \{F_{(i+1)}\}$. The process continues until $i = n$ and $S^n$ contains all of the features from $R$.

Reverse feature addition produces feature sets $S_i$ constructed in an opposite manner to FFA. We start with $S_1 = \{\mathscr{F}_{(n)}\}$ that contains only the lowest ranked feature. The next feature set $S_{i+1}$ is constructed by adding the lowest-ranked feature which is not in $S_i$, namely $S_{i+1} = S_i \cup \{F_{(n-i)}\}$. In the same way as for FFA, the process of RFA continues until we include all of the features, that is, $S_n = \mathscr{F}$.

Note that FFA can be viewed as backward feature elimination. Starting from $S^n = \mathscr{F}$, at each step we remove the least relevant feature from $S^i$ to obtain $S^{i-1}$. Similarly, RFA can be viewed as forward feature elimination. Finally, it holds that $\mathscr{F} = S^{n-i} \cup S_i$ for all $i$.

For each $i$ and each constructed feature subset $S^i$ or $S_i$, we build predictive models $\mathcal{M}(S^i, F_t)$ and $\mathcal{M}(S_i, F_t)$. We then estimate their respective prediction errors, $err^i$ and $err_i$. This results in two error curves. We name them FFA and RFA curves, each constructed by the corresponding FFA/RFA feature subset construction method. The value for each point of the FFA curve is defined as $FFA(i) = err^i$, while for the RFA curve as $RFA(i) = err_i$. The process of FFA/RFA curve construction is summarized in Algorithm 1.

The computational complexity of the proposed algorithm for constructing a single (FFA or RFA) curve is $\mathcal{O}(n(M + T))$, where $n$ is the number of features, $M = M(n)$ is the cost of constructing the predictive model and $T = T(n)$ is the cost of its evaluation. It should be noted that $M$ and $T$ are dependent on the specific learning method used for inducing the model and on the procedure used for evaluating it.

Typically, the points $FFA(i)$, $FFA(i + 1)$, $\cdot$, do not differ considerably, for $i$ large enough, since it expected that only a small proportion of the features is relevant when the data is high-dimensional. This means that we can make the algorithm more efficient if we construct the set $S^{i+\delta(i)}$ from the set $S^i$ by including $\delta(i)$ features into it. Analogously, we speed up the construction of the RFA curves.

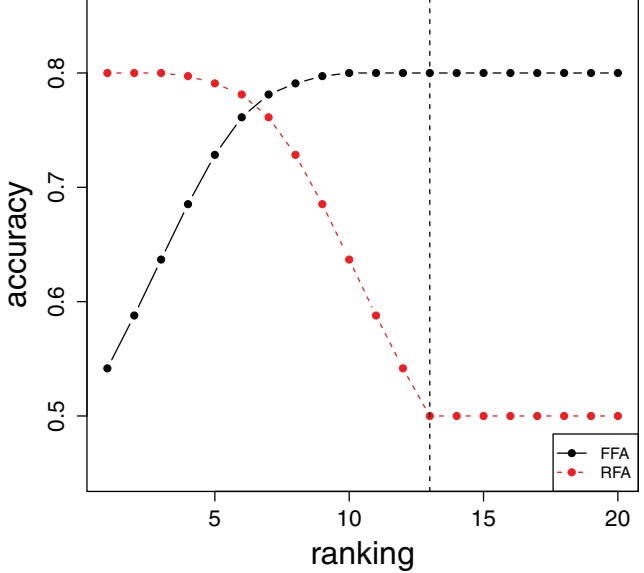

**Figure 1  Sample FFA and RFA curves.**

### Interpretation of error curves

The visualization and interpretation of the FFA and RFA curves can be explained by considering the examples of FFA and RFA curves given in Fig. 1. The $y$-axis for both curves is the same and depicts the error estimate of a feature subset. Point $i$ at the $x$-axis corresponds to the moment when the feature $F_{(i)}$ is first included in the predictive model: $S^i$ for the FFA curve and $S_{n-i+1}$ for the RFA curve. Thus, the FFA curve in Fig. 1 is constructed from left-to-right as the top-ranked features are at the beginning of the ranking. In contrast, the RFA curve is constructed from right-to-left starting with the end of the ranking and going towards its beginning.

Let us first focus on the FFA curve. We can observe that as the number $k$ of features increases, the accuracy of the predictive models also increases. This can be interpreted as follows: By adding more and more of the top-$k$ ranked features, the number of relevant features in the constructed feature subsets increases, which is reflected in the improvement of the accuracy (error) measure.

Next, for the RFA curve in Fig. 1, if we inspect it from right-to-left, we can notice that it is quite different from the FFA curve at the beginning. Namely, the accuracy of the models constructed with the bottom ranked features is minimal, which means the ranking is correct in the sense that it puts only irrelevant features at the bottom of the ranking. As the number of bottom-$k$ features increases, some relevant features are included and the accuracy of the models increases.

We now consider the complete Fig. 1. The FFA and RFA curve essentially provide an estimate of how the relevant features are spread throughout the feature ranking. Namely, the FFA curve provides us with an estimate of where the relevant features appear at the top of the ranking, while the RFA curve provides an estimate of where relevant

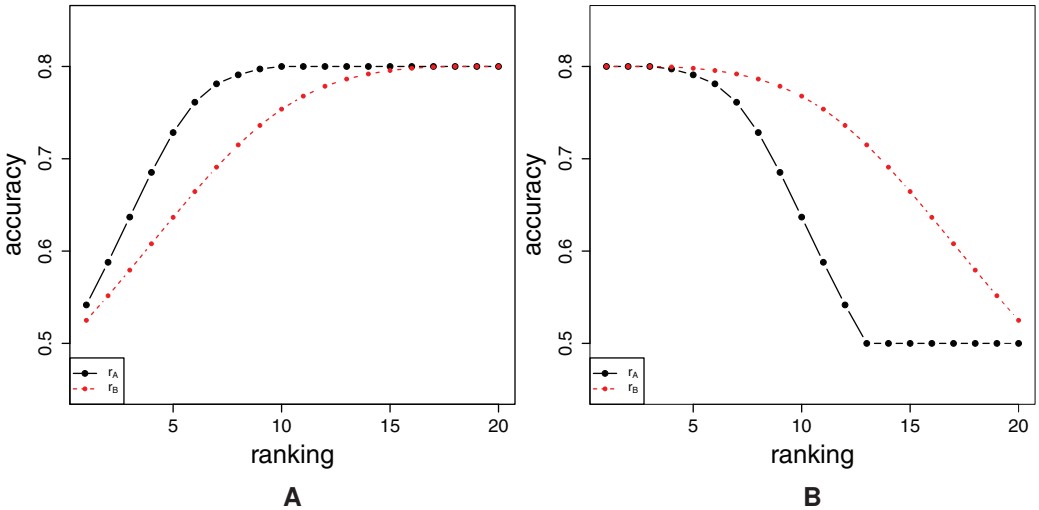

**Figure 2  Comparison of FFA curves (A) and RFA curves (B) of two ranking methods $r_A$ and $r_B$.**

features appear at the bottom of the ranking. In the specific case depicted in Fig. 1, the relevant features are located between the 1st and the 13th ranked feature.

Besides providing an estimate of the spread of the relevant features across the feature ranking, the real utility of the FFA/RFA curves becomes apparent if we consider them in a relative, or more precisely, a comparative context. Let us consider two arbitrary feature ranking methods $r_A$ and $r_B$, which produce feature rankings $\mathbf{R}_A$ and $\mathbf{R}_B$, respectively. For these two rankings, we present the corresponding FFA/RFA curves in Fig. 2.

We first inspect the FFA curves visually. We find that the values of the FFA curve of the ranking method $r_A$ are (most of the time) above the FFA curve of the other ranking method $r_B$. This can be interpreted in the following way: for an arbitrary $k$, when considering the top-$k$ features of the feature rankings $\mathbf{R}_A$ and $\mathbf{R}_B$, more relevant features are included in the top-$k$ features of ranking $\mathbf{R}_A$ than the top-$k$ features of ranking $\mathbf{R}_B$. This implies that ranking algorithm $r_A$ produces a better ranking as compared to the ranking algorithm $r_B$.

A similar discussion applies to the RFA curve. When one considers the bottom-$k$ features of a given feature ranking, most of the time, feature ranking $\mathbf{R}_A$ includes less relevant features than feature ranking $\mathbf{R}_B$, that is, the predictive models constructed are less accurate. Here, because the opposite logic of the FFA curve applies, one can also conclude that the feature ranking algorithm $r_A$ produces a better feature ranking than the feature ranking algorithm $r_B$.

## Expected FFA and RFA curves

When one wants to assess the quality of a single feature ranking in a real-world application, its forward (reverse) feature addition curves can be only compared to the curves that belong the ranking, generated uniformly at random, since the ground-truth

ranking is not known. As discussed before, the random ranking $\mathbf{R}_{RND}$ is the worst-case ranking, since it contains no information about the distribution of the relevant features. As such, it can also serve as a baseline.

The expected values of the points that define FFA curve of the ranking $\mathbf{R}_{RND}$ coincide with the expected values of the RFA curve of this ranking, since the corresponding values only depend on the data itself and the number of features $i$ at a given point of the curves. Thus, *expected curves* can be the common name for both types of the curves that belong to $\mathbf{R}_{RND}$. Computing the exact average error estimations $\mathbb{E}_{\mathbf{S}}[err^i] = \mathbb{E}_{\mathbf{S}}[err_i]$, where $\mathbf{S} \subseteq \mathscr{F}, |\mathbf{S}| = i$, may be unfeasible if the number of features $n$ is large (e.g., for $i = n/2$, $\mathcal{O}((2n)!/(n!)^2)$ models have to be evaluated), but one can overcome this by sampling the sets $\mathbf{S}$.

## Stability of feature ranking

An important aspect of feature ranking algorithms is their stability (*Nogueira, Sechidis & Brown, 2017*) or, more specifically, the stability of the ranked feature lists that they produce. Once we have the set $\mathscr{R}$ of $m$ rankings $\mathbf{R}_t$ that were induced from the different samples of the dataset $\mathscr{D}$, the stability index $S(\mathscr{R})$ can be calculated as

$$St(\mathscr{R}) = \frac{1}{\binom{m}{2}} \sum_{t=1}^{m-1} \sum_{s=t+1}^{m} S_M(\mathbf{R}_t, \mathbf{R}_s),$$

that is, the stability index is the average of pairwise similarities $S_M$ for each pair of rankings. In general, the function $S_M$ can be any (dis)similarity measure, for example, the Spearman rank correlation coefficient (*Saeys, Abeel & De Peer, 2008*; *Khoshgoftaar et al., 2013*), the Jaccard distance (*Saeys, Abeel & De Peer, 2008*; *Kalousis, Prados & Hilario, 2007*), an adaptation of the Tanimoto distance (*Kalousis, Prados & Hilario, 2007*), Fuzzy (Goodman and Kruskal's) gamma coefficient (*Boucheham & Batouche, 2014*, *Henzgen & Hüllermeier, 2015*), etc.

To assess the stability of feature ranking in our experimental work, we set $S_M = \text{Ca}$, where Ca is the Canberra distance (*Lance & Williams, 1966*; *Lance & Williams, 1967*; *Jurman et al., 2008*). This is a weighted distance metric that puts bigger emphasis on the stability of the top ranked features. If we have two feature rankings $\mathbf{R}_A$ and $\mathbf{R}_B$ of $n$ features, then the Canberra distance is defined as

$$\text{Ca}(\mathbf{R}_A, \mathbf{R}_B) = \sum_{j=1}^{n} \frac{|\text{rank}_A(F_j) - \text{rank}_B(F_j)|}{\text{rank}_A(F_j) + \text{rank}_B(F_j)}. \tag{1}$$

However, we do not only estimate the stability of the ranking as a whole. Rather, we also estimate the stability of the partial rankings based on the features from $\mathbf{S}^i$. In order for the distance to be applicable to such partial rankings with $i < n$ features, the following adaptation is proposed: instead of using the ranks $\text{rank}_{A,\ B}(F)$, we use

$\text{rank}_{A,B}^i(F) = \min\{\text{rank}_{A,B}(F), i+1\}$, that is, all features with rank higher than $i$ are treated as if they had rank $i + 1$:

$$Ca(\mathbf{R}_A, \mathbf{R}_B) = \sum_{j=1}^{n} \frac{|\text{rank}_A^i(F_j) - \text{rank}_B^i(F_j)|}{\text{rank}_A^i(F_j) + \text{rank}_B^i(F_j)}. \qquad (2)$$

Additionally, we would like the stability indicator to be independent of specific values of $i$ and $n$. Hence, we normalize it by the expected Canberra distance between random rankings, denoted by $Ca(n, i)$. It can be approximated (*Jurman et al., 2008*) as

$$Ca(n, i) \approx \frac{(i+1)(2n-i)}{n}\log 4 + \frac{i(1+i)}{n} + 2i - 3, \qquad (3)$$

which we make use of when $i \geq 8$ and the computation of the exact value becomes intractable. For $i \geq 8$, the relative error of the approximation (3) is smaller than 1%. Our final stability indicator is thus the curve consisting of points calculated as

$$\left(i, \frac{St(\mathbf{S}^i)}{Ca(n, i)}\right), \qquad (4)$$

for $1 \leq i \leq n$, that represent the relative change of distance between top-$i$ lists w.r.t. the expected top-$i$ distance.

## EMPIRICAL EVALUATION OF FFA/RFA CURVE PROPERTIES

We start with the experiments on synthetic datasets. In such laboratory conditions, one has full control over the data, can establish the ground truth feature ranking, and produce rankings of different quality. Such a setting will facilitate proper assessment of our proposed feature ranking evaluation method. Before proceeding to the experiments, we briefly describe the constructed synthetic datasets. The detailed description of the datasets is given in "Appendix A1".

We construct three datasets named `single`, `pair` and `combined`. Each of them consists of 1,000 instances and 100 features. The relevant features in `single` dataset are individually correlated to target, the relevant features in the `pair` dataset are related to the target via XOR relation, and the relevant features in the `combined` dataset are the union of the relevant features in the `single` and `pair` datasets. The rest of the features in the datasets are random noise.

### Evaluation by randomising the ground truth ranking

The appropriateness of the proposed method is first demonstrated on a family of feature rankings that contain more and more noise. By doing that, we can show that the lower and lower quality of feature rankings is reflected in the FFA and RFA curves, and thus detected by the method.

We start with the ground-truth ranking $\mathbf{R}_{GT}$ and perturb it as follows. First, we select a proportion $\theta$ of randomly selected features which are then assigned random relevances, drawn uniformly from the unit interval. The other features preserve their ground truth relevance. This results in a ranking $\mathbf{R}_\theta$.

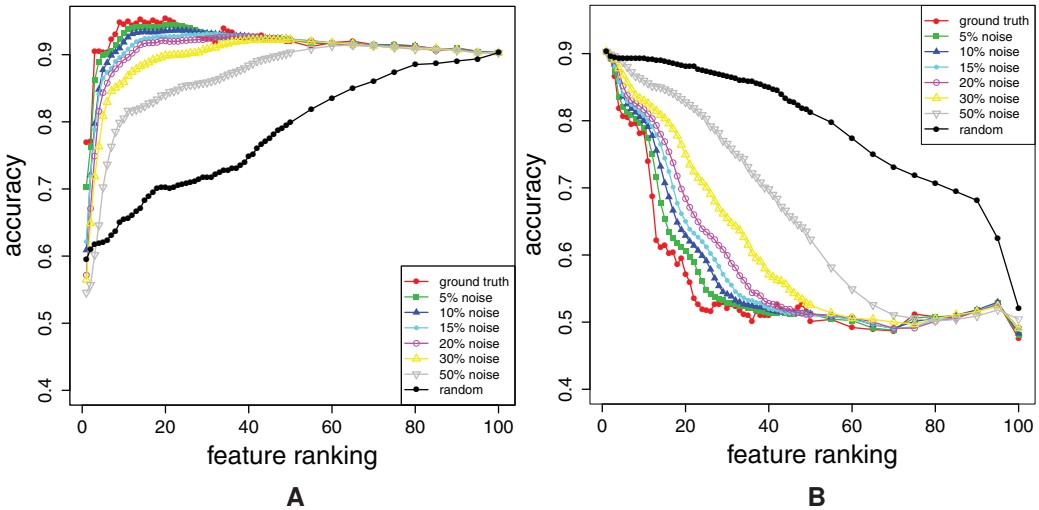

**Figure 3 Dataset combined: Forward feature addition curves (A), and reverse feature addition curves (B).** The curves for the noisy rankings $R_\theta$ ($0.05 \leq q \leq 1$) and the ground truth ranking are shown.

### Experimental setup

We use the aforementioned `single`, `pair` and `combined` datasets. The following amounts $\theta$ of noise are introduced into the ground truth ranking: $\theta \in \{0.05, 0.1, 0.15, 0.2, 0.3, 0.5, 1\}$. The value $\theta = 1$ corresponds to a completely random ranking.

For every value of $\theta$, we estimate the expected values of the FFA/RFA curves that belong to the ranking $\mathbf{R}_\theta$, by first generating $m = 100$ realizations of the ranking, and second, (point-wise) averaging of the error estimates of the obtained predictive models.

For constructing FFA/RFA curves, SVMs were used, as noted and justified at the end of the "Analysis of Different Learning Methods to Construct FFA and RFA Curves". The curves were constructed via 10-times stratified 10-fold cross validation, using different random seeds.

### Results

The obtained FFA and RFA curves are shown in Fig. 3 that gives the results for the dataset `combined`. The results for the datasets `single` and `pair` are similar. In addition to the curves that belong to the rankings $R_\theta$ with different amounts of noise, ground truth ranking is also shown.

Both, FFA curves (Fig. 3A) and RFA curves (Fig. 3B) correctly detect different amount of noise $\theta$: the higher the $\theta$, the more distant are FFA and RFA curve of $R_\theta$ to the curves of ground truth ranking. The independent confirmation of these results are given in "Appendix A2".

Additionally, note that FFA curves cannot give all the information about the ranking: Had we not plotted the RFA curves in Fig. 3B, we would not have a proof that all of the rankings misplace some relevant features (check the considerable decrease in accuracy just before the 100th feature).

### Analysis of different learning methods to construct FFA and RFA curves

According to Algorithm 1, the error curve estimates depend not only on the feature ranking method, but also on the learning method used to construct the predictive models. In this section, we investigate which learning methods (learners) are suitable to construct the FFA and RFA curves. Note that we are not searching for a learner that would produce the most accurate predictive models. Rather, the requirement for the learner to be used in this context is that it should produce predictive models that exploit all the information that the features contain about the target concept, and can thus distinguish between feature rankings of different quality.

#### *Experimental setup*

When comparing the FFA and RFA curves of different ranking methods, constructed with different learners, we used the `combined` dataset described in detail in "Appendix 1". We consider the following four feature ranking methods.

**Information gain**, where we calculate the information gain of each feature $F_i$ as $IG(F_t, F_i) = H(F_t) - H(F_t : amp:mid; F_i)$.

**SVM-RFE** uses a series of support vector machines (SVMs) to recursively compute a feature ranking. A linear SVM was employed, as proposed by (*Guyon et al., 2002*). Following (*Slavkov et al., 2018*), we set $\varepsilon = 10^{-12}$ and $c = 0.1$.

**ReliefF** algorithm as proposed by *Robnik-Šikonja & Kononenko (2003)*. The number of neighbors was set to 10, and the number of iterations was set to 100%.

**Random Forests**, which can be used for estimating feature relevance as described by *Breiman (2001)*. A forest of 100 trees was used, constructed by randomly choosing rounded up $\log_2$ of the number of features.

We compare the above ranking methods by using different learners to produce classifiers and generate error estimates for the FFA and RFA curves. More specifically, we consider **Naïve Bayes** (*John & Langley, 1995*); **Decision Trees** (*Quinlan, 1993*); **Random Forests** (*Breiman, 2001*): the number of trees was set to 100, and in each split $\log_2$ of the number of features are considered; **SVMs** (*Cortes & Vapnik, 1995*): a polynomial (quadratic) kernel was employed, with the $\varepsilon = 10^{-12}$ and $c = 0.1$; **k-NN** (*Aha & Kibler, 1991*) with a value of $k = 10$.

The curves were constructed via 10-times stratified 10-fold cross validation, using different random seeds. The obtained FFA and RFA curve comparisons of the four feature ranking methods obtained by each of the five learning methods, are presented in the following section.

#### *Results*

The rankings are shown in Fig. 4, where each graph represents the distribution of the ground truth relevance values. The *y*-axis depicts the ground truth relevance value (5). Each point, *i*, represents the *i*-th ranked feature, $F_{(i)}$, as determined by the feature ranking method.

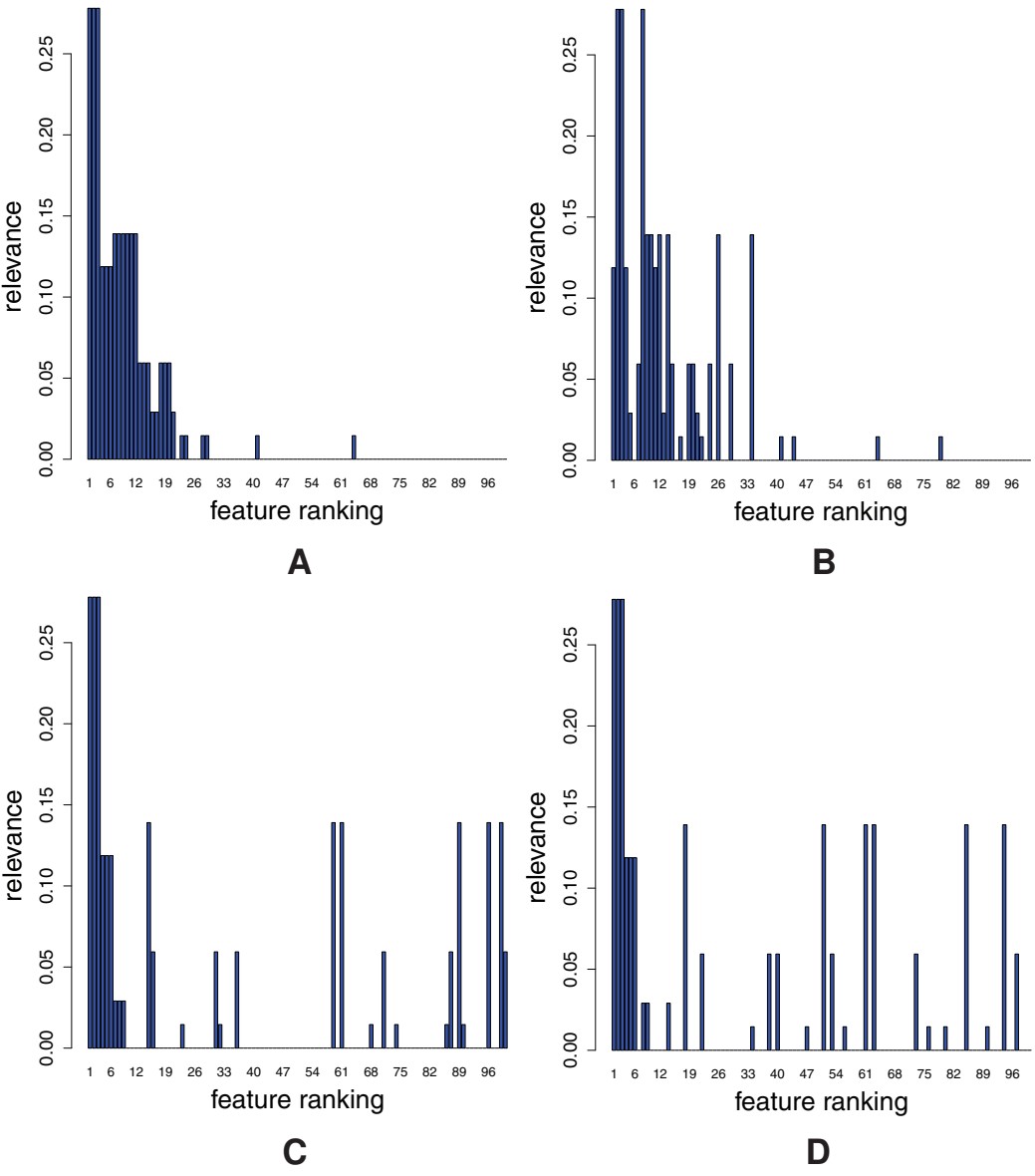

**Figure 4 Distribution of relevant features for each of the four ranking methods: ReliefF (A), Random Forests (B), Info Gain (C) and SVM-RFE (D).**

We can see that the rankings fall into two groups: in Figs. 4A and 4B, highly relevant features are concentrated on the left, while in the Figs. 4C and 4D, they are evenly spread.

ReliefF and Random Forests (Figs. 4A and 4B) are thus clearly better than Info Gain and SVM-RFE (Figs. 4C and 4D). Hence, the FFA and RFA curves should at least differentiate between the two groups of the rankings. However, there should be visible difference also between Relief and Random Forests at the beginning of the ranking. The detailed comparative evaluation of the obtained feature rankings is given in "Appendix A3".

In the case of FFA curves, the learners can be divided into two groups: FFA curves produced by Naïve Bayes, Decision Trees and Random Forests cannot capture any
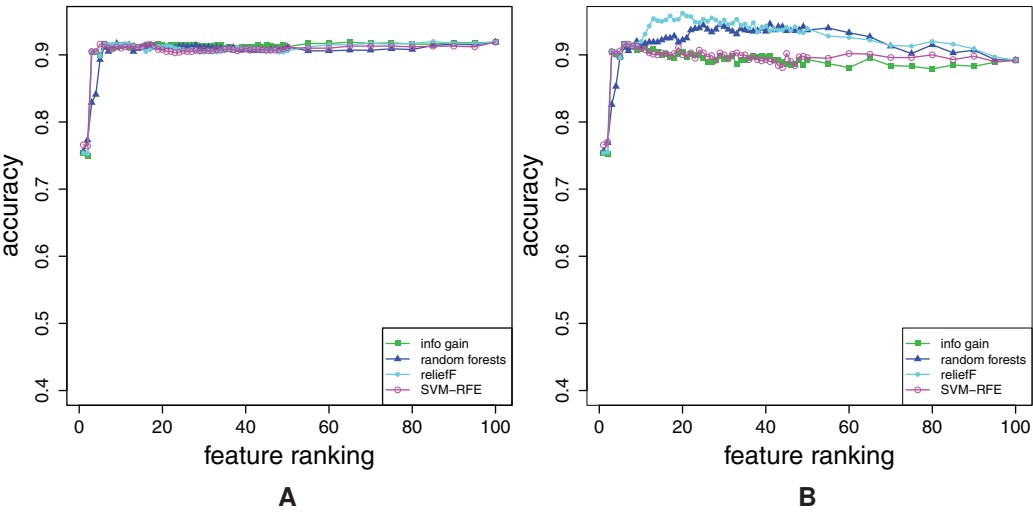

**Figure 5 Comparison of FFA curves for the four different ranking methods for the `combined` dataset.** The curves were obtained by using the Naïve Bayes (A) and k-NN (B).

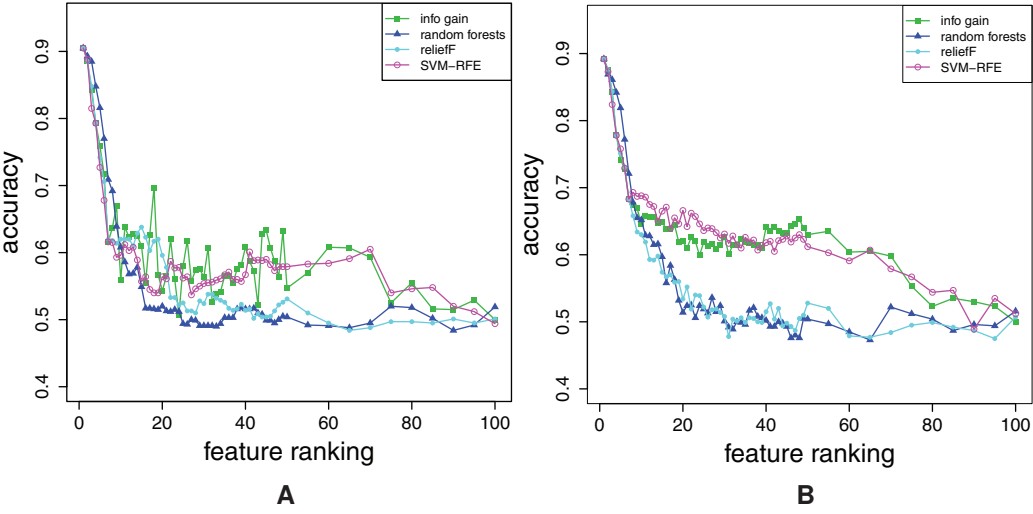

**Figure 6 Comparison of RFA curves for the four different ranking methods for the `combined` dataset.** The curves were obtained by using Decision Trees (A) and k-NN (B).

difference between rankings, whereas those produced by SVMs and k-NN can. It suffices to show one representative graph for each group (those for Naïve Bayes and k-NN are shown in Fig. 5), since there are no significant differences among the learning methods in the same group[1]. The FFA curves produced by these two learners have all the desirable properties: the curves for Relief and Random Forests are better than those of Info Gain and SVM-RFE. Additionally, at the beginning, the Random Forest curve is under the curve of Relief.

[1] Compare, for example, Fig. 5B (obtained by k-NN) with Fig. A1A (obtained by SVMs), which is given in Appendix 3 (note that Fig. A1A also contains the random ranking curve).

The reason why, for example, the Naïve Bayes classifier does not show any difference between rankings, is the fact that it can not use the information from the interactions of higher order. Namely, it assumes feature independence. Hence, it is not appropriate for use in the considered context.

If we proceed to RFA curves, again, the Naïve Bayes classifier does not show any difference between rankings, whereas the other four methods do. We prefer Random Forests, SVMs and k-NN over Decision Trees in the case of RFA curves, because Decision Trees generate quite unstable curves, as shown in Fig. 6A. In Fig. 6B, the RFA curves of k-NN are shown. Again, there is no quantitative difference between them and the RFA curves generated by SVMs[2].

Tu sum up, one can use

- SVMs and k-NN models, for constructing FFA curves,
- SVMs, k-NN and Random Forests, for constructing RFA curves.

Thus, only k-NN and SVMs are appropriate for constructing both FFA and RFA curves. Since one should typically use approximate k-NN when the number of features is extremely high (*Muja & Lowe, 2009*), we use SVMs (with the settings described here) as the learner for constructing the FFA/RFA curves in all the remaining experiments in this work.

### Discussion

We have to give some additional notes about choosing the best method, when, for example, different learning methods prioritize different rankings, which is possible since some learning method might make use of some features, whereas another learning method can make better use of some others.

If we have computed feature rankings to learn a classifier that uses only a subset of (top-ranked) features and we have already decided on which classifier to use, we should use the same (type of) classifier to construct the curves, because we want to use the features that the chosen learning method prioritizes.

Second, if our motivation for computing the feature rankings is to discover all relevant features for a given problem (e.g., the genes that influence the patients' clinical state), and learning method $A$ prioritizes the ranking $\mathbf{R} = (x_1, x_2, \ldots)$ over the ranking $\mathbf{R}' = (x_{1'}, x_{2'}, \ldots)$, whereas learning method $B$ prioritizes $\mathbf{R}'$ over $\mathbf{R}$, this means that $x_1, x_2, x_1'$ and $x_2'$ are important (provided that both learners achieve similar accuracy), so we can include them all in the subsequent experiments (and thus use both learning methods).

The decision about which among the two appropriate methods to use—k-NN or SVMs—might also depend on the properties of the dataset at hand. As mentioned before, k-NN could be too time-consuming when the number of features is extremely high. On the other hand, if the number of instances is high, SVMs could be too time-consuming, but speed-ups are possible (*Tsang, Kwok & Cheung, 2005*). As for the noise, both methods are quite robust (*Wang, Jha & Chaudhuri, 2018*; *Xu, Caramanis & Mannor, 2009*), so this is not among the most influential factors.

[2] Compare Figs. 6B–A1B (given in "Appendix A3").

# FEATURE RANKING COMPARISON ON REAL-WORLDS DATASETS

In this section, we move from the synthetic data and show the appropriateness of the proposed feature ranking evaluation method on the real-world data with unknown relevant and irrelevant features. To be consistent with the synthetic-data experiments, we evaluate the same four feature ranking methods as before, and compare them to the random feature rankings which now serve as the only baseline.

## Datasets description

In this extensive study, 35 classification benchmark problems are used. They come in two bigger groups. The first group has been part of the experiments in (*Slavkov et al., 2018*) and except for `aapc` (*Džeroski et al., 1997*), `water` and `diversity` (*Džeroski, Demšar & Grbović, 2000*), mostly originates from the UCI data repository (*Newman & Merz, 1998*). This benchmark problems have higher number of instances (up to 5,000) and not extremely high number of features (up to 280). This problems cover various domains: health, ecology, banking etc.

The second group is newly included and contains 11 high-dimensional micro-array benchmark problems (*Mramor et al., 2007*) (up to 12,625 features) with lower number of examples (up to 110). The main properties of the data are given in Table 1.

## Experimental setup

We construct the curves that base on the feature ranking methods described in experimental setup part of "Analysis of Different Learning Methods to Construct FFA and RFA Curves", and the curves that belong to the completely random ranking (i.e., expected curves) which serve as a baseline. For the actual construction of the curves (once the ranking is obtained), support vector machines were used, as described and justified in the results in "Analysis of Different Learning Methods to Construct FFA and RFA Curves". The curves were constructed via 10-times stratified 10-fold cross validation, using different random seeds.

The expected error curves for random rankings were produced by generating 100 random rankings for each dataset under consideration. For each random ranking, error curves were produced and the average of the error values was used as the expected error. This was done in a manner similar to the one described in "Evaluation by Randomising the Ground Truth Ranking". As mentioned in "The Evaluation Method", building FFA/RFA curves by adding the features one by one to large feature subsets $S^i$ and $S_i$, might be too costly when $n$ is big enough. In this set of experiments, we use the following procedure. We add $\delta(i)$ features to the subset, where $\delta(i)$ is defined as follows: $\delta(i) = 1$ if $1 \leq i \leq 50$, $\delta(i) = 5$ if $50 < i \leq 500$, and $\delta(i) = n//20$ otherwise, where $//$ denotes integer division.

## Results

In this section, we show representative examples of three types of curves: FFA, RFA and stability curves. The curves are shown for two datasets with lower and two with higher

**Table 1 Properties of the benchmark datasets: number of instances, number of features, number of discrete/numeric attributes, and number of different class values.**

| Dataset | #Inst. | #Feat. | (D/N) | #Cl. |
|---|---|---|---|---|
| aapc | 335 | 84 | (83/1) | 3 |
| arrhythmia | 452 | 280 | (73/206) | 16 |
| australian | 690 | 14 | (8/6) | 2 |
| balance | 625 | 4 | (0/4) | 3 |
| breast-cancer | 286 | 9 | (9/0) | 2 |
| breast-w | 699 | 9 | (9/0) | 2 |
| car | 1,728 | 6 | (6/0) | 4 |
| chess | 3,196 | 36 | (36/0) | 2 |
| diabetes | 768 | 8 | (0/8) | 2 |
| diversity | 292 | 86 | (0/86) | 5 |
| german | 1,000 | 20 | (13/7) | 2 |
| heart | 270 | 13 | (6/7) | 2 |
| heart-c | 303 | 13 | (7/6) | 5 |
| heart-h | 294 | 13 | (7/6) | 5 |
| hepatitis | 155 | 19 | (13/6) | 2 |
| image | 2,310 | 19 | (0/19) | 7 |
| ionosphere | 351 | 34 | (0/34) | 2 |
| iris | 150 | 4 | (0/4) | 3 |
| sonar | 208 | 60 | (0/60) | 2 |
| tic-tac-toe | 958 | 9 | (9/0) | 2 |
| vote | 435 | 16 | (16/0) | 2 |
| water | 292 | 80 | (0/80) | 5 |
| waveform | 5,000 | 21 | (0/21) | 3 |
| wine | 178 | 13 | (0/13) | 3 |
| amlPrognosis | 54 | 12,625 | (0/12,625) | 2 |
| bladderCancer | 40 | 5,724 | (0/5,724) | 3 |
| breastCancer | 24 | 12,625 | (0/12,625) | 2 |
| childhoodAll | 110 | 8,280 | (0/8,280) | 2 |
| cmlTreatment | 28 | 12,625 | (0/12,625) | 2 |
| colon | 62 | 2,000 | (0/2,000) | 2 |
| dlbcl | 77 | 7,070 | (0/7,070) | 2 |
| leukemia | 72 | 5,147 | (0/5,147) | 2 |
| mll | 72 | 12,533 | (0/12,533) | 3 |
| prostate | 102 | 12,533 | (0/12,533) | 2 |
| srbct | 83 | 2,308 | (0/2,308) | 4 |

**Note:**
The datasets with a considerably high number of features are listed under the dashed line.

number of features. The graphs for the other datasets can be found in "Appendix 4" in Figs. A2–A12.

We start with the `breast-w` dataset. The FFA/RFA curves in Figs. 7A and 7B show that both types of curves are needed in order to evaluate the ranking completely. The FFA-

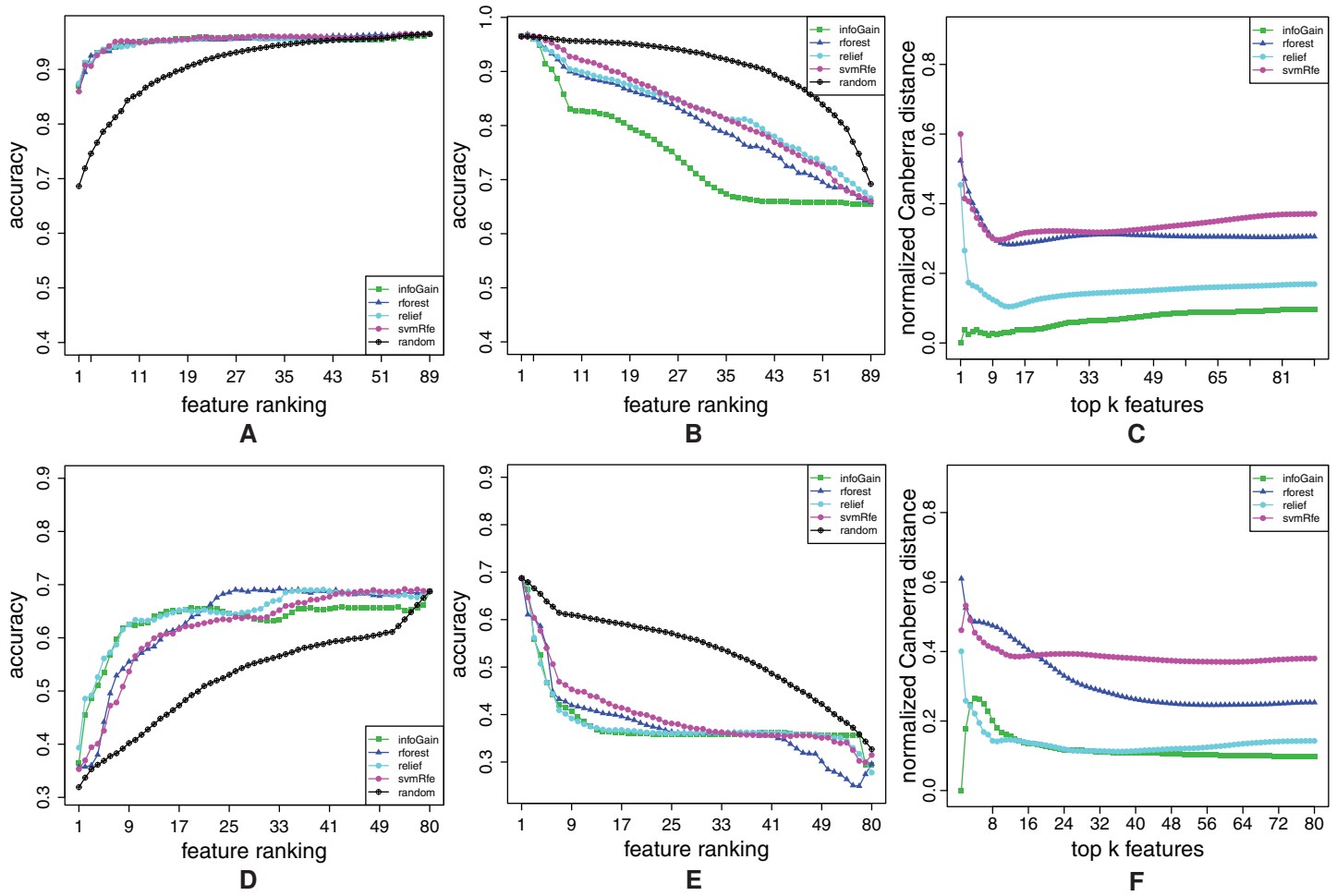

**Figure 7 Ranking quality assessment for datasets `breast-w` (A–C) and `water` (D–F) in terms of the FFA (A and D) and RFA curves (B and E), and rankings' stability estimates (C and F).** The FFA/RFA curves are obtained by using SVMs.

curves suggest that all feature ranking algorithms (except for the random ranking) place only the relevant features at the beginning, since there is practically no difference if we compare the accuracy of the 89-feature (all) model and, for example, the 11-feature model. However, the RFA-curves show that all feature ranking algorithms—except for Info Gain—misplace some relevant features, since the Info-Gain-ranking-based models have the lowest accuracy by far in the RFA curves. Also, in the case of Info Gain, the accuracy cease to decrease after only cca. A total of 40 top-ranked features were removed.

Figure 7C shows that Info Gain produces also the most stable rankings. We can see that the top-ranked feature is always the same, since the stability index of the Info Gain equals 0 at the point $k = 1$. The second most stable algorithm is ReliefF, the third is Random Forest and the least stable is SVM-RFE, but the difference between Random Forest and SVM-RFE is not considerable.

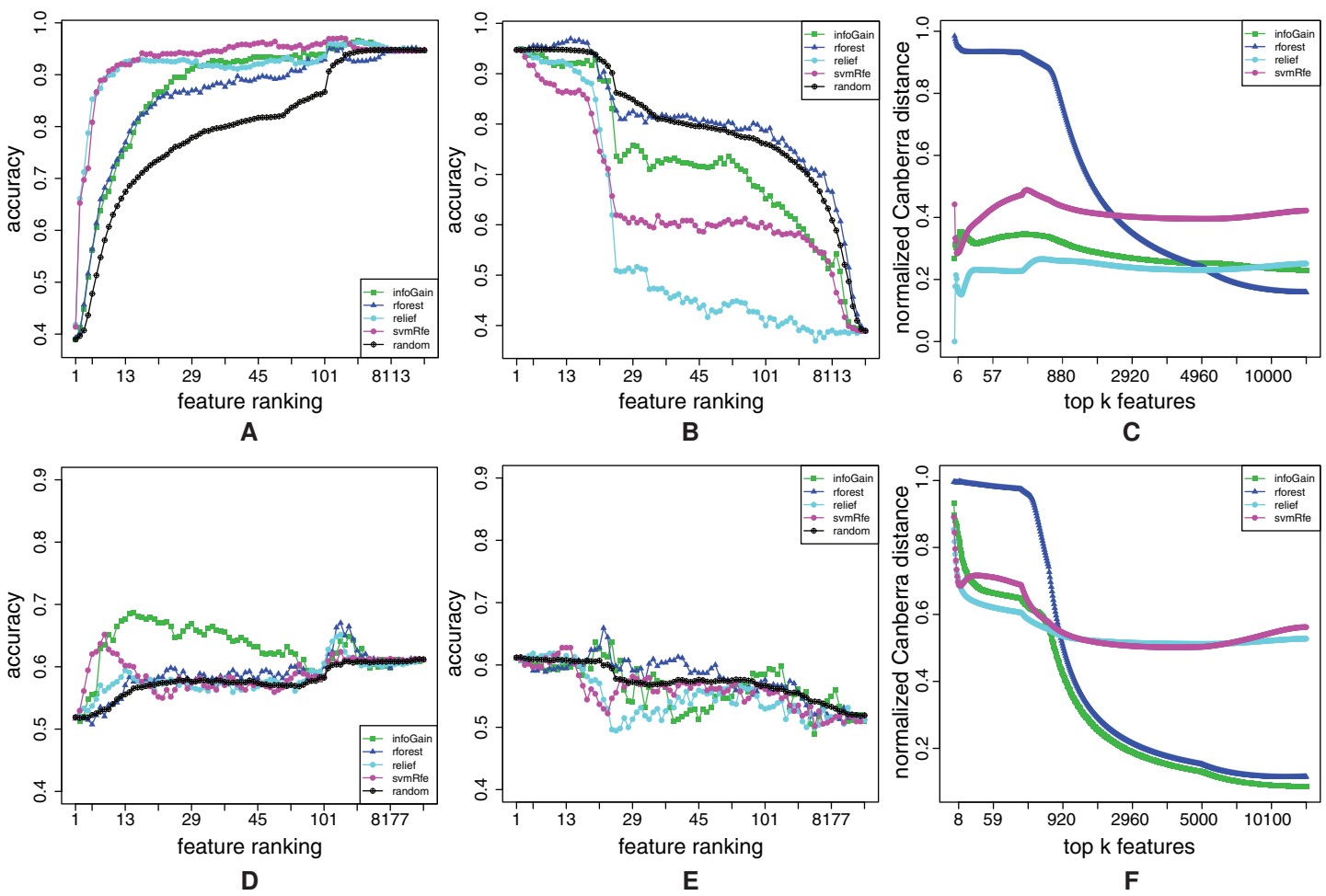

**Figure 8 Ranking quality assessment for datasets `mll` (A–C) and `amlPrognosis` (D–F) in terms of the FFA (A and D) and RFA curves (B and E), and rankings' stability estimates (C and F).** The FFA/RFA curves are obtained by using SVMs.

Let us now take a look at the curves for the dataset water. From the FFA-curves in Fig. 7D, we see that ReliefF, Info Gain and Random Forest ought to have the same 21 top-ranked features, and as a consequence, the same last 59 = 80 − 21 features too. However, the first 21 features are ordered better by Info Gain and ReliefF, while the last 59 are more properly ordered by Random Forest. We can conclude that none of the rankings is ideal, but we can come close to the ideal one (in terms of FFA-curves), if we combine the first part of the ReliefF (or Info Gain) and the second part of Random Forest. This claim is also confirmed by the RFA-curves of Info Gain and ReliefF (Fig. 7E): these two algorithms indeed misplace some relevant features, since the accuracy of the model abruptly decreases and the end of the ranking.

Figure 7F suggests that we should prefer Info Gain and ReliefF over Random Forest since they are more stable. However, we can also notice that Random Forest is the least stable at the beginning of the ranking but its stability increases when the number of features gets larger.

We begin the analysis of the high-dimensional datasets with `mll`. Figure 8B shows that Random Forest completely misplaces some relevant features, since its RFA-curve mostly goes above the random-ranking one. Even though it is evident from Random Forest's FFA-curves that some relevant features are also successfully captured, Random Forest produces the worst ranking. Info Gain is slightly better, whereas ReliefF and SVM-RFE are again the best algorithms. From the FFA-curves, we can conclude that SVM-RFE places more features with higher relevance at the beginning of the ranking (its curve is higher than ReliefF's), while RFA-curves reveal that SVM-RFE also misses some quite relevant features: ReliefF's curve is far below SVM-RFE's. Figure 8C shows that ReliefF is considerably more stable than SVM-RFE, hence we prefer the former over the latter on the `mll` dataset.

The last example will show that sometimes, the understanding of the results is not that easy. In the Figs. 8D and 8E, the FFA and RFA curves for the `amlPrognosis` dataset are presented. In this case, only Info Gain performs considerably better than random ranking in terms of FFA-curves. SVM-RFE is also able to find some relevant features at the beginning (peak of its curve at 10 features), but after that, the models' accuracy decreases, hence mostly noisy features are positioned here. Some relevant features are again placed by ReliefF, Info Gain and Random Forest also around the 2,000th place (local peak of their curves in the right part of the FFA-curve). RFA-curves confirm that there is indeed much noise in these data, since removing features does not result in an (at least approximately) decreasing curve.

It may not come as a surprise that all ranking algorithms produce rankings that are very unstable at the beginning (Fig. 8F), but it is interesting that after approximately 1,000 features, Info Gain and Random Forest produce quite stable rankings even though they have low quality. The reason for both low quality of rankings and their instability is probably the low number of instances accompanied by a high number of features (54 and 12,625 respectively).

## CONCLUSIONS

We have proposed a method for evaluating and comparing feature rankings. The method is based on constructing two chains of predictive models that are built on two families of nested subsets of features. The first family of subsets are the sets of top-ranked features, while the second family consists of sets of bottom-ranked features. The predictive performances of the models form a FFA curve in the former case, and RFA curve in the latter case.

We show in our experiments that both types of curves are necessary when comparing the rankings: FFA curves detect whether important features are placed at the beginning of the ranking, whereas RFA curves detect whether important features can still be found at the end of the ranking.

In the set of experiments, we show the usefulness of the proposed evaluation method and its sensitivity to the rankings of different quality on synthetic data. The second set of experiments shows which of the learning methods are appropriate for building the

FFA and RFA curves (SVMs, k-NN) and which are not (Naïve Bayes, Decision Tree, Random Forest). In the third set of experiments on synthetic data, we test several feature ranking algorithms and examine their properties. Considering data with different properties, we show that ReliefF algorithm outperforms the other investigated approaches, both in terms of detecting relevant features and in terms of stability of the feature rankings it produces.

Moreover, we show the usefulness of the proposed approach in real-world scenarios. We evaluate feature rankings computed by four feature ranking algorithms on 35 classification benchmark problems. The results reveal that there is no feature ranking algorithm that would dominate the others on every dataset.

A possible disadvantage of the proposed method is that it can be computationally quite intensive, if we want to construct the curves in full resolution. Namely, every point of a FFA or RFA curve comes at the cost of building and evaluating a predictive model. However, as justified in the method description, the full resolution is, especially when the number of features is really high, not necessary, and, moreover, the construction of the curve can be also easily parallelized.

The work presented in this article can continue in many directions. First, of all, the proposed methodology could use other error measures, since accuracy is appropriate only for the task of classification when the distribution of target variable is approximately uniform. The strong modularity of the FFA/RFA curves allows for their use in any other predictive modeling task, for example, for the task of regression, we could use root mean squared error instead of accuracy. However, even though there exists a regression version of most of the learners, which are considered for constructing the curves, experiments should be repeated on those cases, since the conclusions about, for example, the most appropriate learner for constructing the curves, can be different. Moreover, the method can be adapted not only to the regression setting, but also to different tasks of structured output prediction (*Bakr et al., 2007*) and time series prediction.

## APPENDIX 1

In this section, we explain how we generate our synthetic datasets. For simplicity, we take both the features $F_i$ and the target $F_t$ to be binary and take values from the set $\{0,1\}$. We then partition the set of features $\mathscr{F}$ into feature interaction subsets $\mathscr{F}_{int}$ of cardinality one and two. The feature sets with cardinality one are single features $F_i$ that are in an individual interaction with the target $F_t$, while the features from the interaction sets with cardinality two determine the value of the target by the *XOR* relation.

The examples are generated as follows. For each example, we first randomly (from a uniform distribution) set the value of the target feature $F_t$. After that, if $\mathscr{F}_{int} = \{F_i\}$, then the value of feature $F_i$ is randomly chosen, so that $P(F_i = F_t) = p$. Otherwise, we have $\mathscr{F}_{int} = \{F_i, F_j\}$, and the values of the features $F_i$ and $F_j$ are randomly chosen, so that $P(XOR(F_i, F_j) = F_t) = p$.

We consider the probability values $p \in \{0.8, 0.7, 0.6, 0.5\}$. The feature sets with $p = 0.5$ are in fact independent of the target $F_t$, and can be considered as irrelevant features.

**Table 2 Properties of the synthetic datasets.**

| $\mathscr{F}_{int}$ | $p$ | #Copies in Single | #Copies in Pair | #Copies in Combined |
|---|---|---|---|---|
| | 0.8 | 3 | | 3 |
| | 0.7 | 3 | | 3 |
| | 0.6 | 3 | | 3 |
| | 0.8 | | 3 | 3 |
| | 0.7 | | 3 | 3 |
| | 0.6 | | 3 | 3 |
| | 0.5 | 91 | 82 | 73 |

Note:
If $p > 0.5$, #copies denotes the number of copies of the interaction set. In the last row where $p = 0.5$, #copies corresponds to the number of independently realized irrelevant features.

With combinations of these feature interaction sets, three datasets were generated, each of them consisting of 1,000 instances and 100 features in total.

The first dataset (named `single`) comprises only individually correlated features. The second dataset (named `pair`) contains relevant features related to the target via the XOR relation, as well as irrelevant features. The third (named `combined`) is a combination of the first two. It contains XOR-related features and individually correlated features.

In order to simulate the redundancy of features, which occurs in real datasets, the three datasets are constructed in the following way: If the set $\mathscr{F}_{int}$ of relevant features is included in the dataset, we also include two redundant copies of $\mathscr{F}_{int}$ in the dataset. Irrelevant features are realized independently of each other.

The properties of the generated datasets are summarized in Table 2, from which we can observe that there are 9 relevant features in the `single` dataset, 18 in the `pair` dataset, and 27 in the `combined` dataset.

The ground-truth feature relevances $rel(F_i)$ of the features $F_i$ are defined as follows. First, the relevance of each feature group $\mathscr{F}_{int}$ is defined as the mutual information between the group and $F_t$, namely $rel(\mathscr{F}_{int}) = I(\mathscr{F}_{int}; F_t)$. Second, for $F_i \in \mathscr{F}_{int}$, feature importances are obtained as

$$rel(F_i) = rel(\mathscr{F}_{int})/|\mathscr{F}_{int}|. \tag{5}$$

For the particular three datasets, this ground-truth ranking $R_{GT}$ should also result in the optimal FFA and RFA curves, but this may not be the case in general. In the next section, we give the results of comparing it to the other feature rankings.

## APPENDIX 2

When discussing the results in the "Evaluation by Randomising the Ground Truth Ranking", we showed that, when the level of noise $\theta$ in the ranking increases, then (i) the quality of the ranking $\mathbf{R}_\theta$ decreases, and (ii) the rankings $\mathbf{R}_{GT}$ and $\mathbf{R}_\theta$ become more and more distant. However, for the second point, we need to define a distance $dist(\mathbf{R}_{GT}, \mathbf{R}_\theta)$ between a noisy and the ground truth ranking. In the definition of $dist(\mathbf{R}_{GT}, \mathbf{R}_\theta)$ we use the average Spearman rank correlation coefficient $\rho(\mathbf{R}_A, \mathbf{R}_B)$ which is calculated as

**Table 3 The distances dist (RGT, Rq) for different $q$ values, for each of the three synthetic datasets.**

| θ | 0.05 | 0.1 | 0.15 | 0.2 | 0.3 | 0.5 | 1 |
|---|---|---|---|---|---|---|---|
| Single | 0.219 | 0.34 | 0.449 | 0.51 | 0.628 | 0.81 | 1.056 |
| Pair | 0.126 | 0.239 | 0.327 | 0.397 | 0.519 | 0.726 | 1.091 |
| Combined | 0.1 | 0.171 | 0.252 | 0.32 | 0.432 | 0.652 | 1.048 |

$$\frac{1}{n-1}\sum_{i=1}^{n}\frac{\left(\text{rank}_A(F_i)-\overline{\text{rank}_A}\right)\left(\text{rank}_B(F_i)-\overline{\text{rank}_B}\right)}{\sigma_A\sigma_B}$$

where $n$ is the number of features, therefore the average ranks $\overline{rank_A}$ and $\overline{rank_B}$ equal $(n+1)/2$. Standard variations $\sigma_{A,B}$ are computed as $\sigma_{A,B}=\sqrt{\sum_{i=1}^{n}\left(\text{rank}_{A,B}(F_i)-\overline{\text{rank}_{A,B}}\right)^2}/(n-1)$. For a given θ, the distance between rankings $\mathbf{R}_{GT}$ and $\mathbf{R}_\theta$ is then computed as

$$dist(\mathbf{R}_{GT},\mathbf{R}_\theta)=1-\frac{1}{m}\sum_{t=1}^{m}\rho(\mathbf{R}_{GT},\mathbf{R}_{\theta,t}) \tag{6}$$

where $m$ is the number of different realizations $\mathbf{R}_{\theta,t}$ of the noisy ranking $\mathbf{R}_\theta$.

Table 3 lists the values of the distances between the ground truth ranking $R_{GT}$ and its noisy versions $\mathbf{R}_\theta$. Note that, for all three synthetic datasets, the distances indeed increase when the amount of noise θ increases.

# APPENDIX 3

In "Evaluation by Randomising the Ground Truth Ranking", we analyzed rankings of different quality (with different amounts of noise) by comparing them to the ground truth ranking. In a real-world setting, the ground truth ranking is unknown and the feature rankings are induced directly from data. Therefore, in this section we analyze feature rankings, produced by the four feature ranking methods from "Analysis of Different Learning Methods to Construct FFA and RFA Curves", and the synthetic data described in "Generating Synthetic Data".

When comparing the rankings, stability should also be taken into account as discussed earlier. Therefore, the stability indicator (4) is also included in the analysis.

## Experimental setup

We have used the same parameter settings for the feature ranking algorithms as in "Analysis of Different Learning Methods to Construct FFA and RFA Curves". As noted and justified in the corresponding "Results", SVMs were used for constructing the FFA/RFA curves. The curves were constructed via 10-times stratified 10-fold cross validation, using different random seeds. To complement them, we also estimate the stability of each feature ranking algorithm by using the stability indicator described in "Stability of Feature Ranking". All feature ranking methods were tested only on the combined dataset.

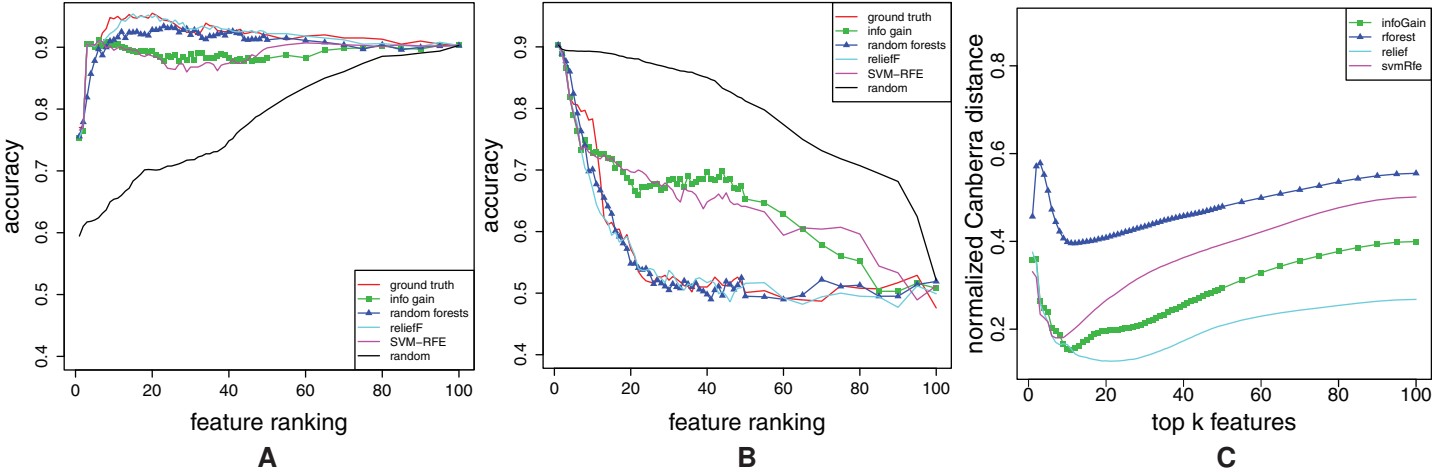

**Figure A1 Ranking quality assessment for dataset `combined` in terms of the FFA (A) and RFA curves (B), and rankings' stability estimates (C).** The FFA/RFA curves are obtained by using SVMs.

## Results

For our analysis, we consider three types of graphs. The first two types are FFA curves (Fig. A1A) and RFA curves (Fig. A1B). The third is the stability estimate graph (Fig. A1C) where the $y$-axis refers to the value of the stability indicator (4): the higher the value, the less stable the ranking method. Each point, $k$, on the $x$-axis represents the size of the considered feature subsets, consisting of the top ranked $k$ features.

Upon a visual inspection of the overall results in Fig. 4, we can conclude that all of the feature ranking methods can correctly detect the features individually related to the target. However, Info Gain and SVM-RFE (Figs. 4C and 4D, respectively) exhibit random behavior for the XOR features, that is, are unable to assign proper relevance values to them. Random Forests (Fig. 4B) separate relevant from irrelevant features, but the ordering of the relevant features is mixed. Finally, ReliefF (Fig. 4A) provides the ranking that is the closest to the ground truth.

These differences in behavior among the different ranking methods are also clearly reflected in the FFA and RFA curves in Figs. A1A and A1B. In Fig. A1B, the RFA curves for Info Gain and SVM-RFE have a similar behavior: Namely, a linearly increasing accuracy (from right to left) in the region where the relevant features are randomly distributed and a sharp increase in accuracy in the region where the individually relevant features are included. On the other hand, the RFA curves of both random forests and ReliefF remain first constant and then increase abruptly when the top-ranked features are included. These two groups of methods can be also distinguished from the FFA curves. The FFA curves of all methods are first increasing abruptly and then slightly decreasing but the FFA curves of ReliefF and random forests increase during more steps and reach higher accuracy than Info Gain and SVM-RFE. This clearly indicates that while Info Gain and SVM manage to identify a proportion of the relevant features and put them at the

top of the ranking, this proportion is nevertheless smaller than the one identified by random forests and ReliefF.

Forward feature addition and RFA curves undoubtedly allow us to compare the quality of the different ranking algorithms. The FFA/RFA curves of all methods are clearly better than the curves of the random ranking. The ReliefF ranking algorithm, however, clearly outperforms the other methods. It has the best error curves, for example, the curves that are the closest to the ground truth ranking. The second best method are random forests: they exhibit very similar performances, but show a slightly worse FFA curve. Both Info Gain and SVM-RFE are clearly inferior in terms of both FFA and RFA curves.

Stability-wise, as seen in Fig. A1C, all of the algorithms are stable in the region of the relevant features that they can detect, except for random forests, which has an instability peak exactly in this region. This means that random forests are in this case capable of detecting all the relevant features, but are highly unstable in the estimation of their ordering. Further inspection reveals that Relief generates not only the best rankings in terms of FFA/RFA curves, but also the most stable ones.

## APPENDIX 4

In this section, we provide detailed per dataset results from the experimental study.

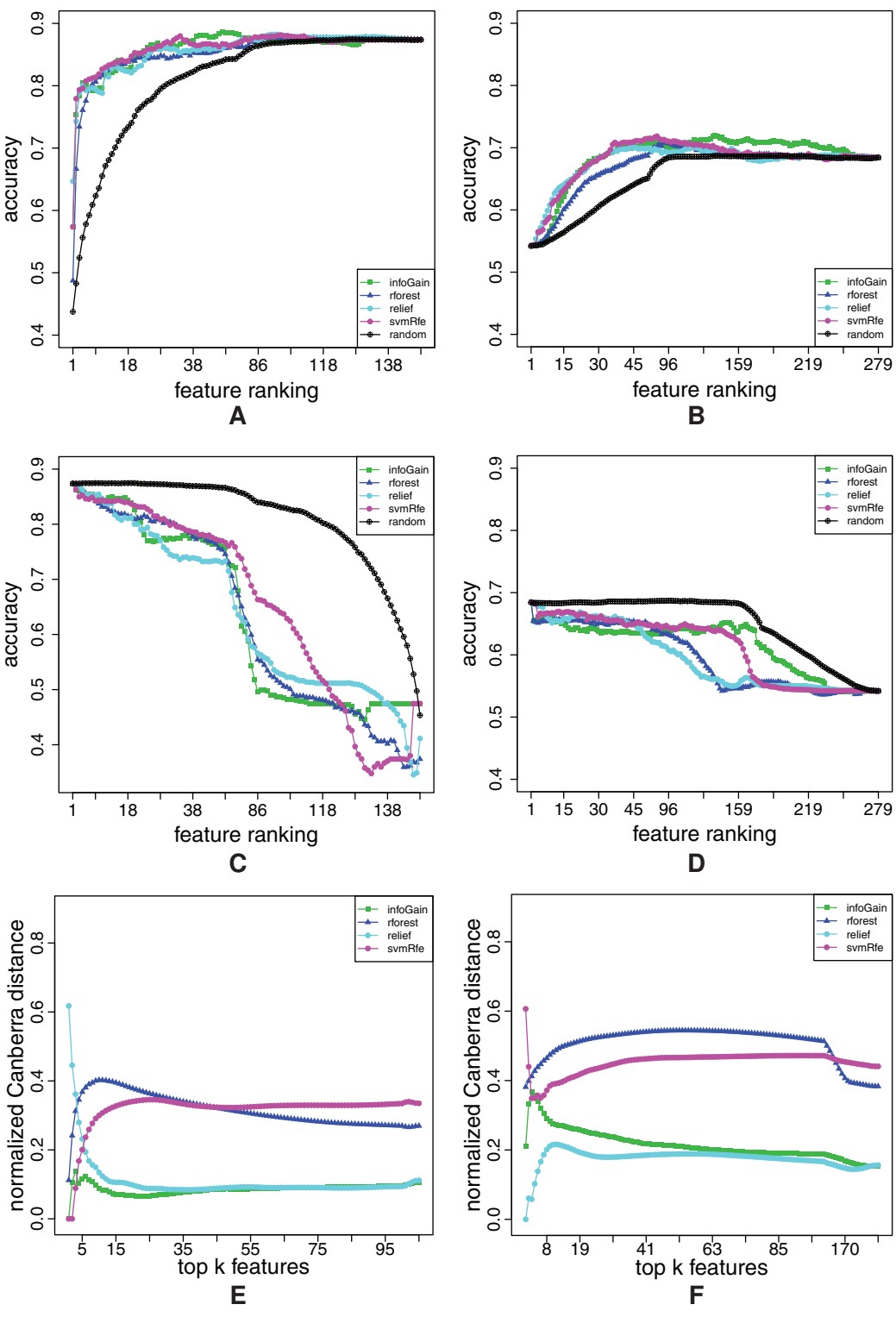

**Figure A2 Ranking quality assessment for datasets `aapc` (A, C and E) and `arrhythmia` (B, D and F) in terms of the FFA (A and B) and RFA curves (C and D), and rankings' stability estimates (E and F).** The FFA/RFA curves are obtained by using SVMs.

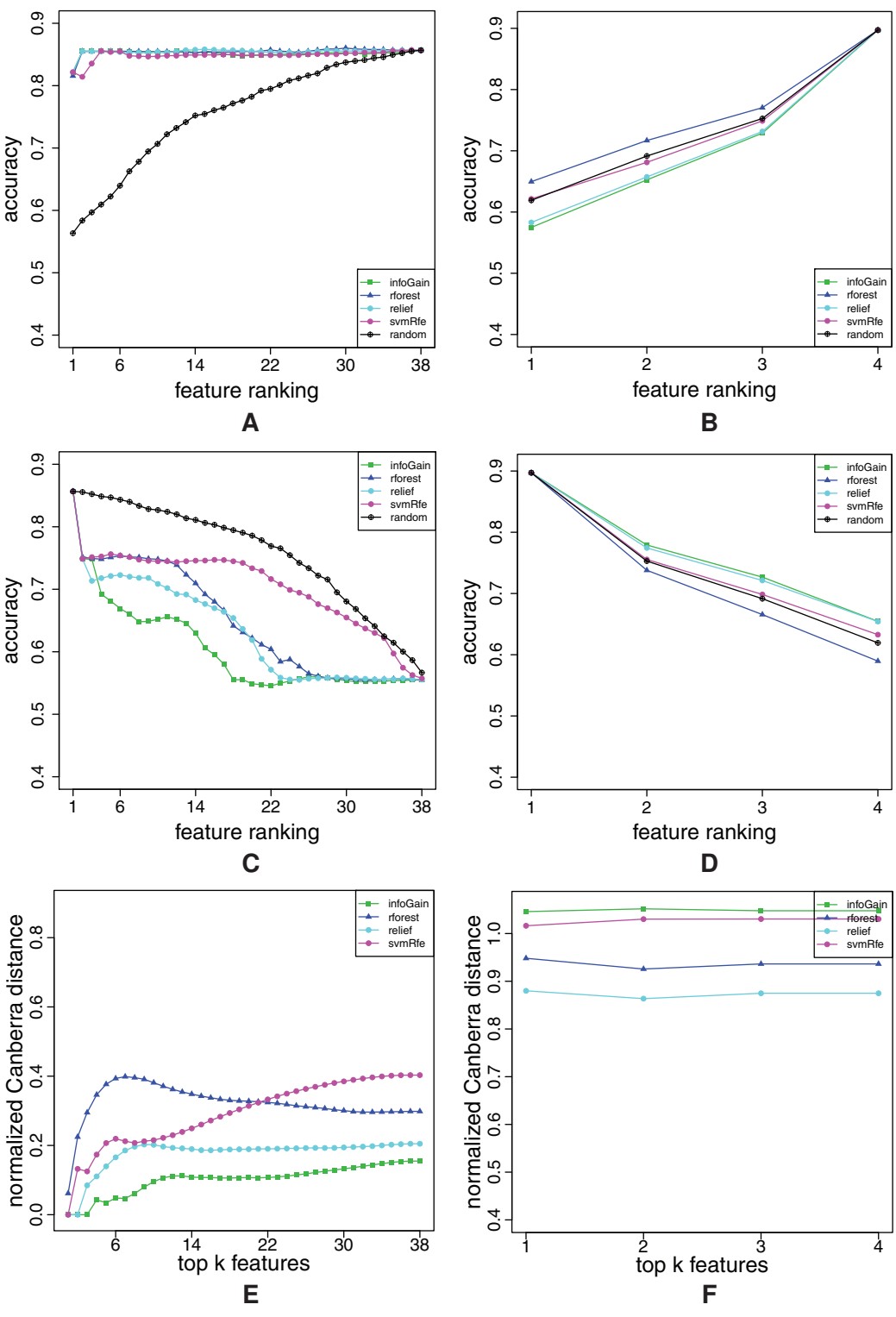

**Figure A3** **Ranking quality assessment for datasets `australian` (A, C and E) and `balance` (B, D and F) in terms of the FFA (A and B) and RFA curves (C and D), and rankings' stability estimates (E and F).** The FFA/RFA curves are obtained by using SVMs.

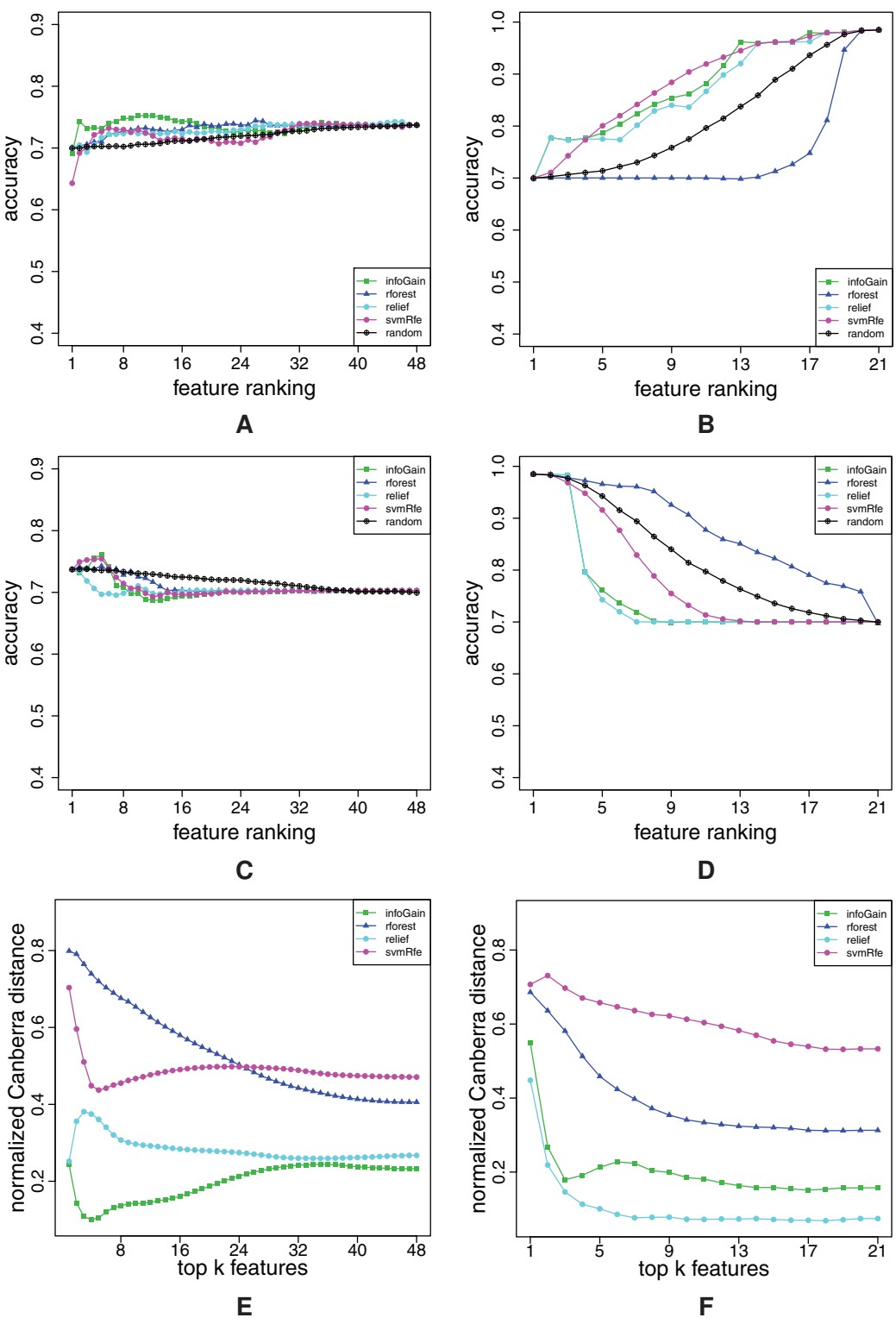

**Figure A4 Ranking quality assessment for datasets `breast-cancer` (A, C and E) and `car` (B, D and F) in terms of the FFA (A and B) and RFA curves (C and D), and rankings' stability estimates (E and F).** The FFA/RFA curves are obtained by using SVMs.

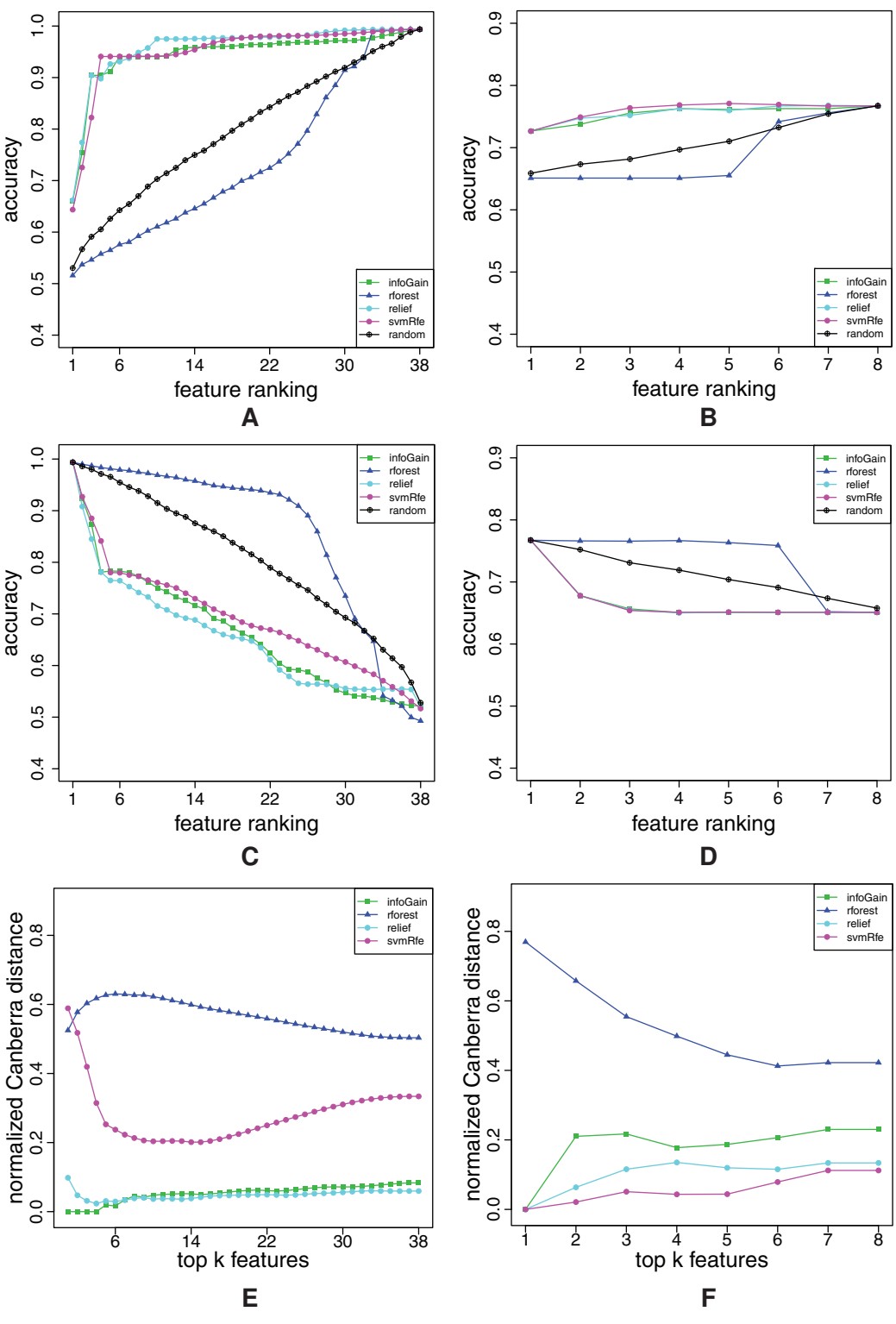

**Figure A5** Ranking quality assessment for datasets `chess` (A, C and E) and `diabetes` (B, D and F) in terms of the FFA (A and B) and RFA curves (C and D), and rankings' stability estimates (E and F). The FFA/RFA curves are obtained by using SVMs.

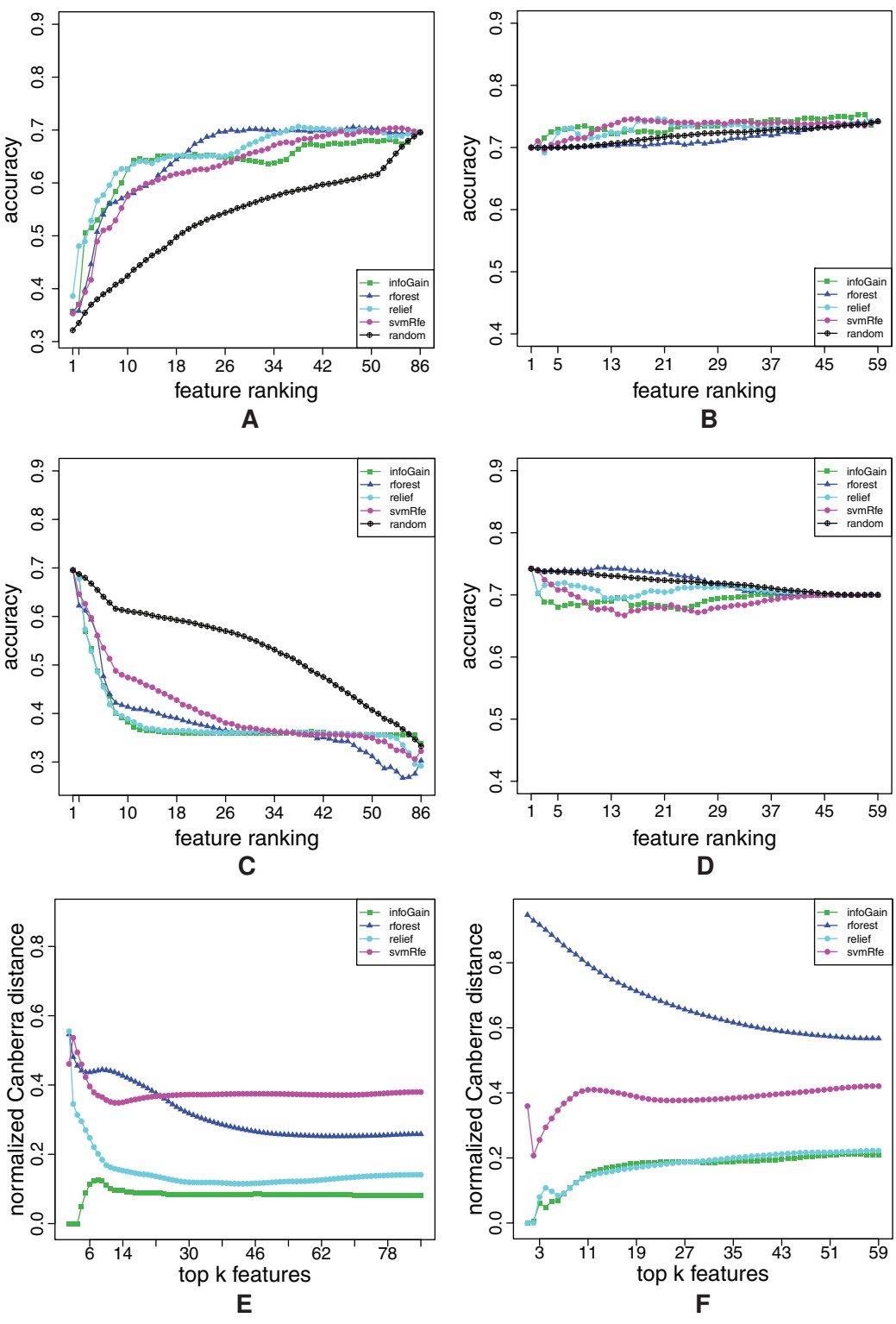

**Figure A6 Ranking quality assessment for datasets `diversity` (A, C and E) and `german` (B, D and F) in terms of the FFA (A and B) and RFA curves (C and D), and rankings' stability estimates (E and F).** The FFA/RFA curves are obtained by using SVMs.

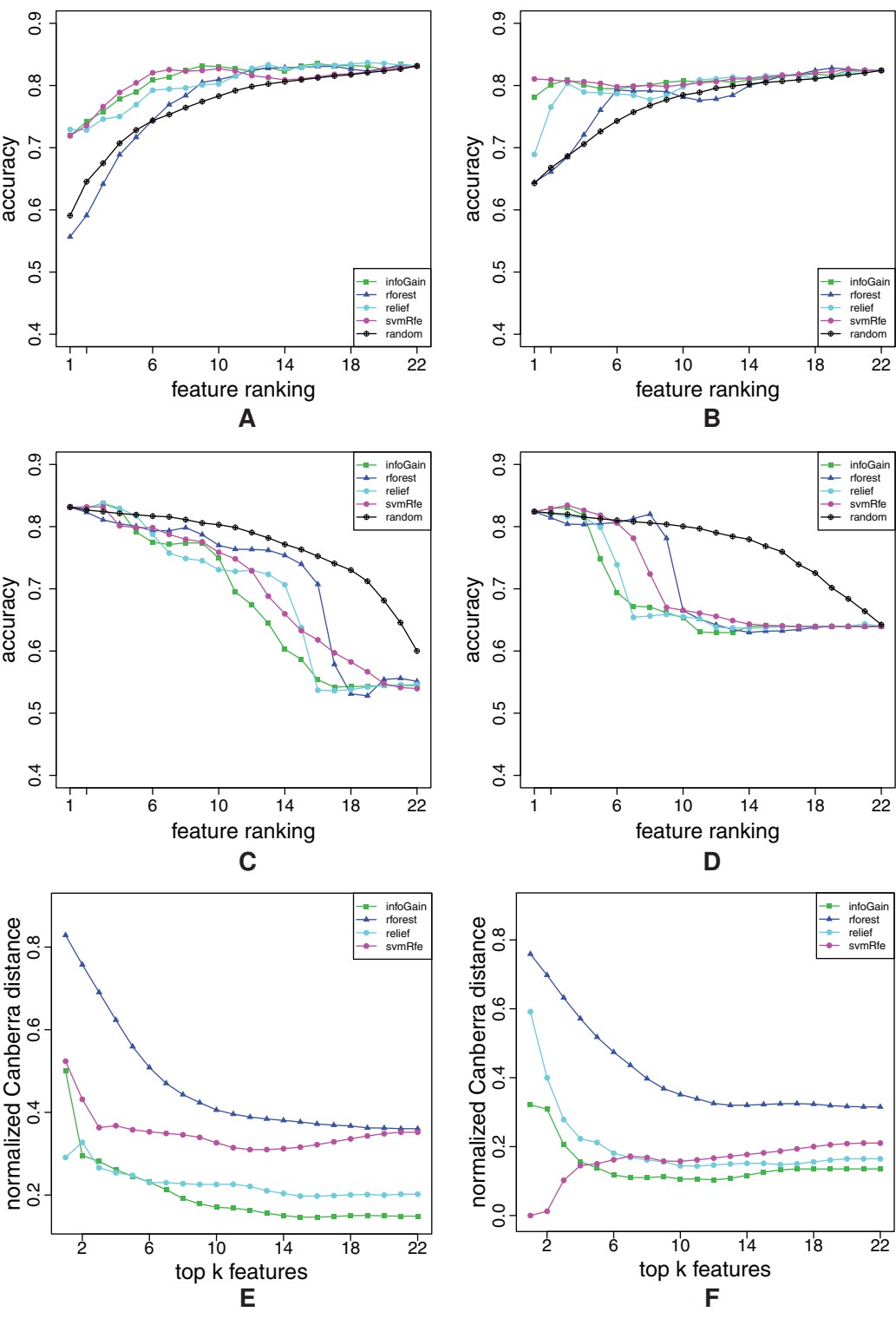

**Figure A7 Ranking quality assessment for datasets `heart-c` (A, C and E) and `heart-h` (B, D and F) in terms of the FFA (A and B) and RFA curves (C and D), and rankings' stability estimates (E and F).** The FFA/RFA curves are obtained by using SVMs.

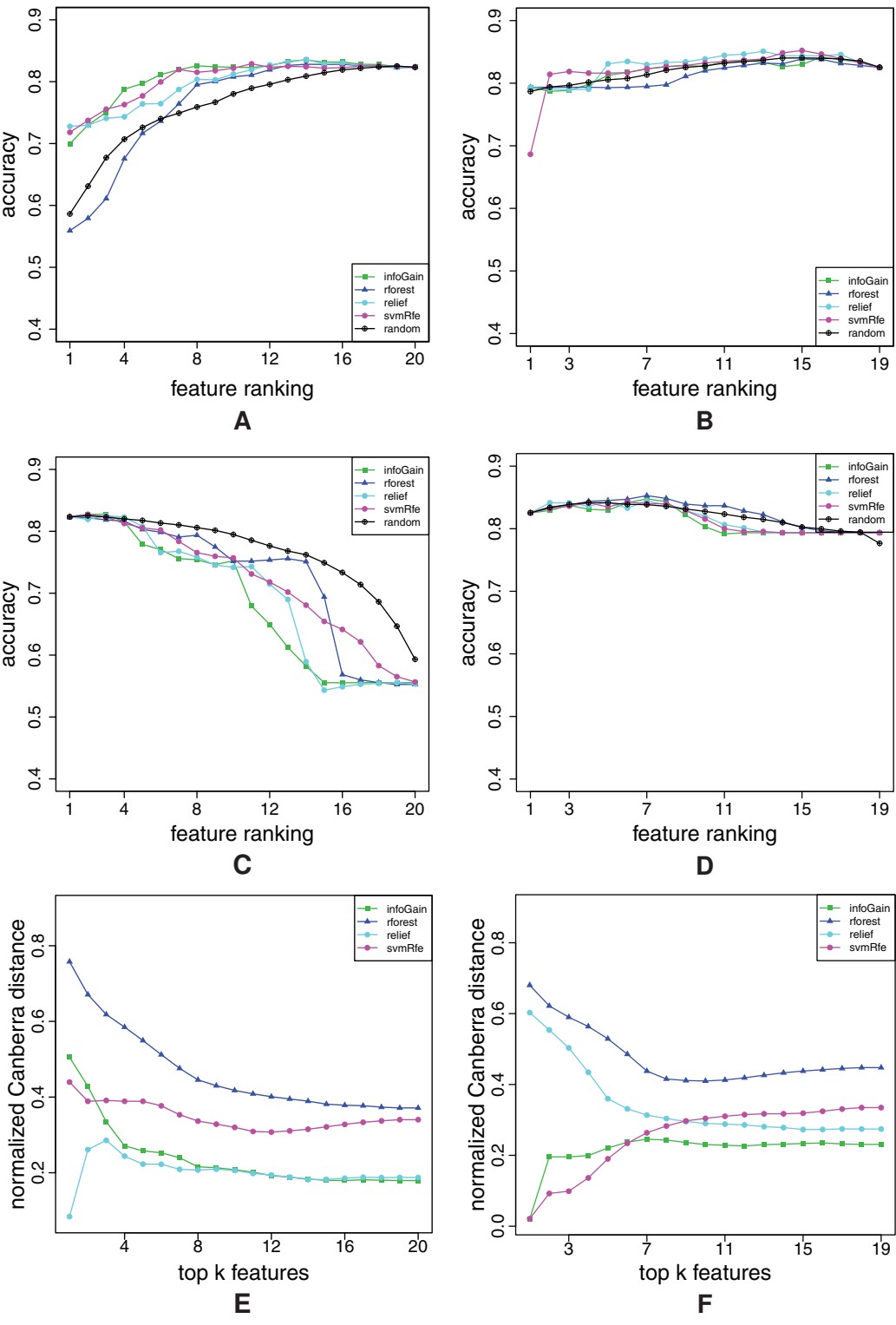

**Figure A8 Ranking quality assessment for datasets `heart` (A, C and E) and `hepatitis` (B, D and F) in terms of the FFA (A and B) and RFA curves (C and D), and rankings' stability estimates (E and F).** The FFA/RFA curves are obtained by using SVMs.

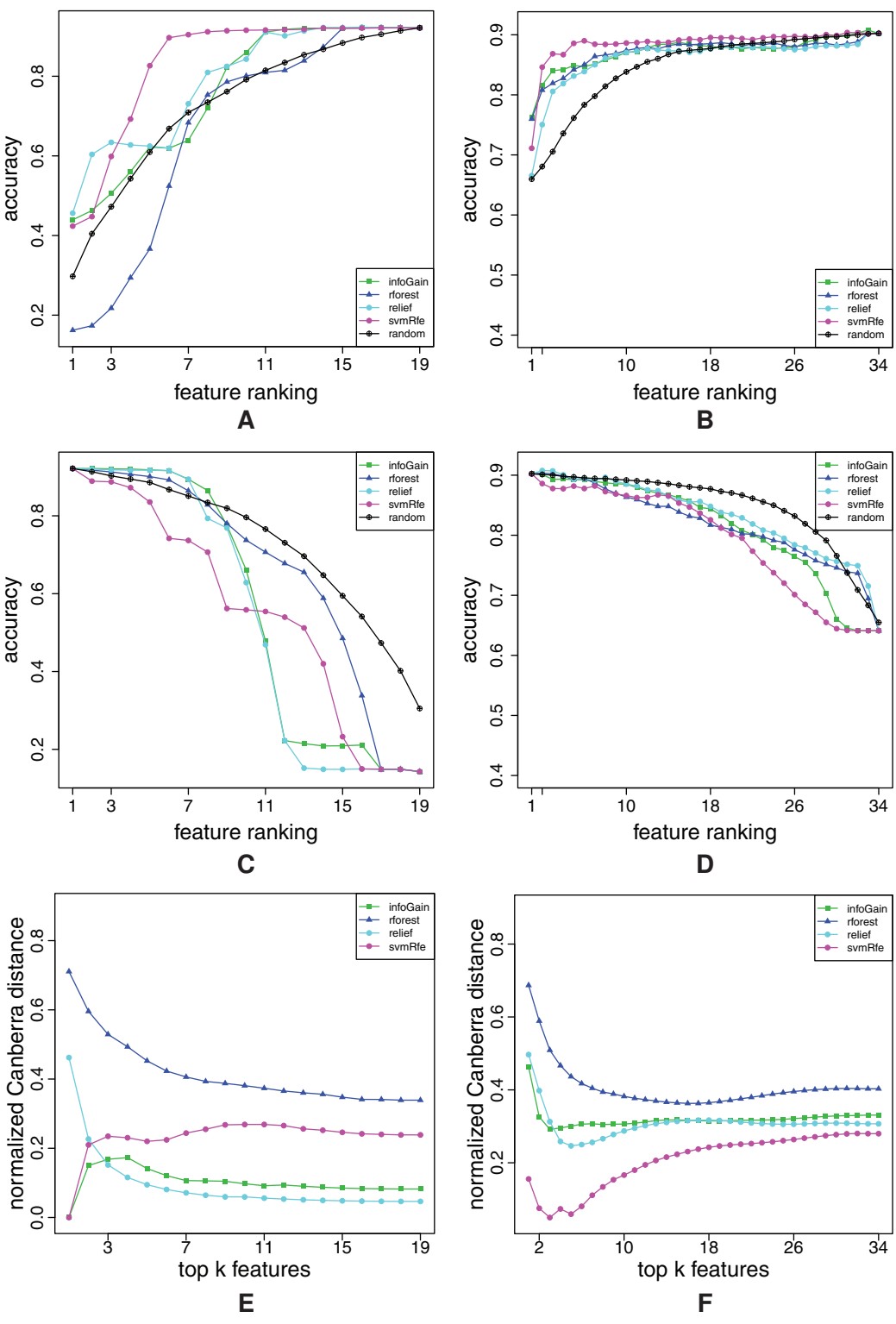

**Figure A9 Ranking quality assessment for datasets `image` (A, C and E) and `ionosphere` (B, D and F) in terms of the FFA (A and B) and RFA curves (C and D), and rankings' stability estimates (E and F).** The FFA/RFA curves are obtained by using SVMs.

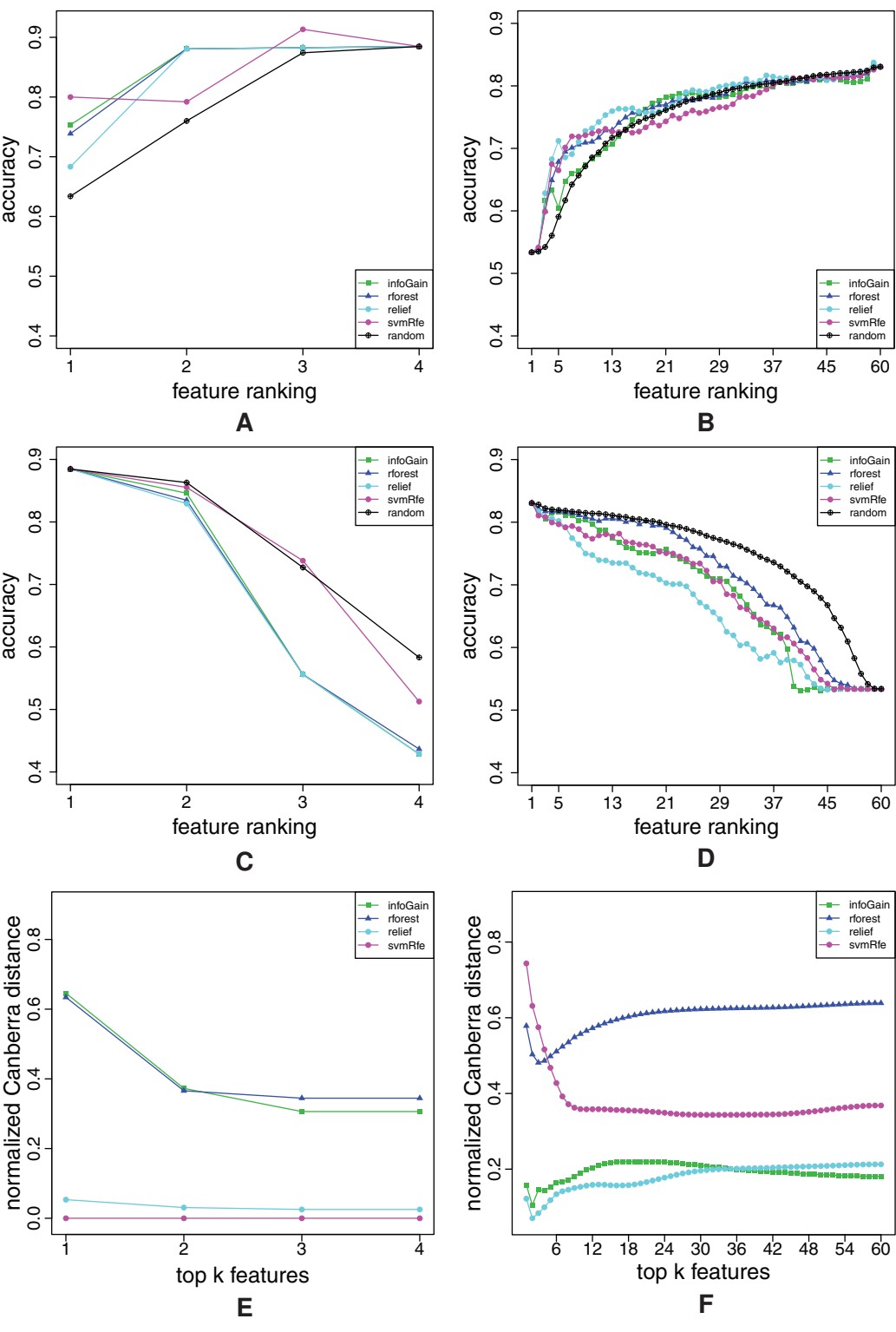

**Figure A10** **Ranking quality assessment for datasets `iris` (A, C and E) and `sonar` (B, D and F) in terms of the FFA (A and B) and RFA curves (C and D), and rankings' stability estimates (E and F).** The FFA/RFA curves are obtained by using SVMs.   

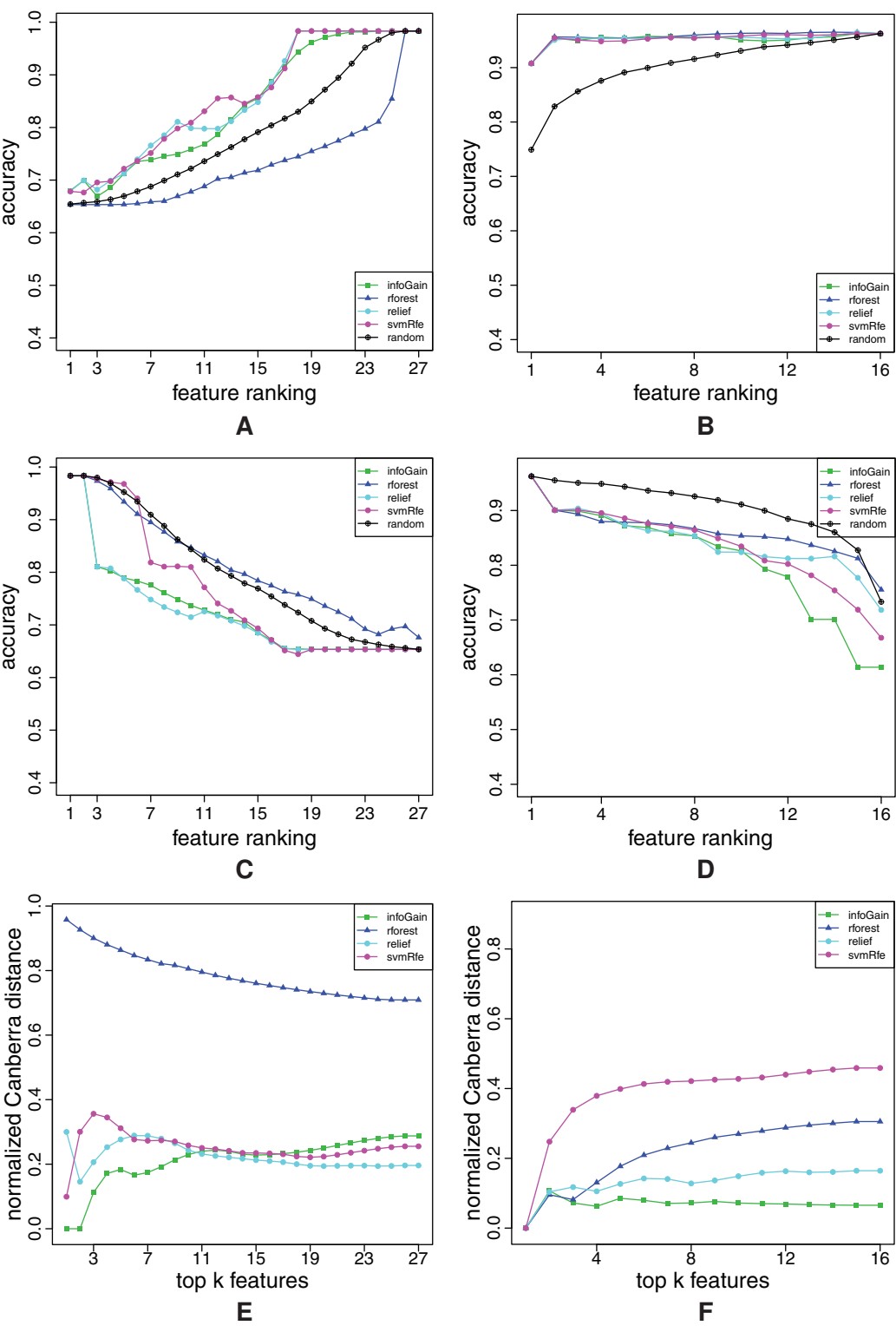

**Figure A11 Ranking quality assessment for datasets `tic-tac-toe` (A, C and E) and `vote` (B, D and F) in terms of the FFA (A and B) and RFA curves (C and D), and rankings' stability estimates (E and F).** The FFA/RFA curves are obtained by using SVMs.

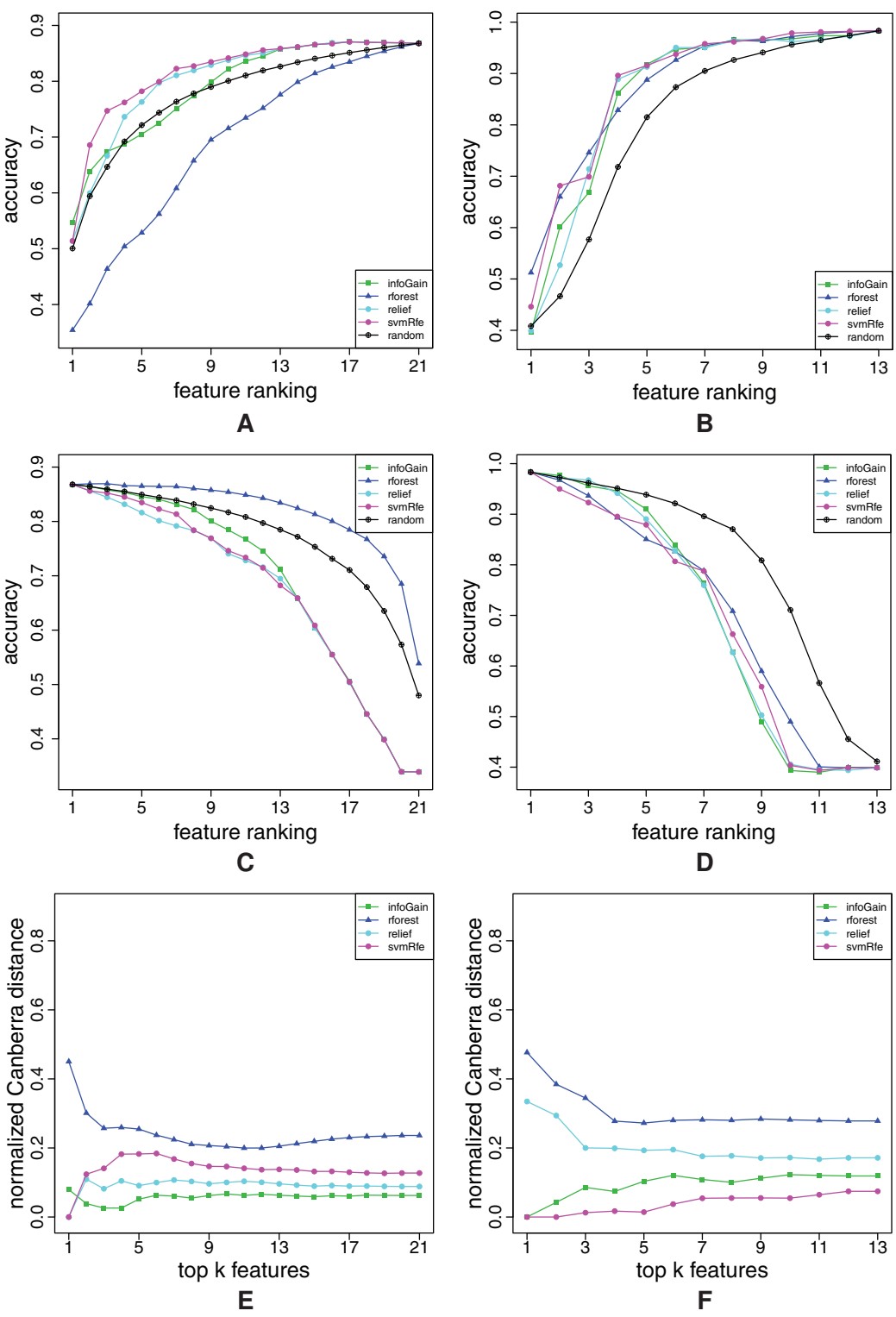

**Figure A12  Ranking quality assessment for datasets `waveform` (A, C and E) and `wine` (B, D and F) in terms of the FFA (A and B) and RFA curves (C and D), and rankings' stability estimates (E and F).** The FFA/RFA curves are obtained by using SVMs.

### Funding

This work was supported by The Ad Futura Slovene Human Resources Development and Scholarship Fund, Slovenian Research Agency (through the grants J2-9230 and N2-0128 and a young researcher grant), the European Commission through the grants TAILOR (H2020-ICT-952215) and AI4EU (H2020-ICT-825619). The funders had no role in study design, data collection and analysis, decision to publish, or preparation of the manuscript.

### Grant Disclosures

The following grant information was disclosed by the authors:
The Ad Futura Slovene Human Resources Development and Scholarship Fund, Slovenian Research Agency: J2-9230 and N2-0128.
TAILOR (H2020-ICT-952215) and AI4EU (H2020-ICT-825619).

### Competing Interests

The authors declare that they have no competing interests.

### Author Contributions

- Ivica Slavkov conceived and designed the experiments, performed the experiments, analyzed the data, performed the computation work, prepared figures and/or tables, authored or reviewed drafts of the paper, and approved the final draft.
- Matej Petković performed the experiments, analyzed the data, performed the computation work, prepared figures and/or tables, authored or reviewed drafts of the paper, and approved the final draft.
- Pierre Geurts conceived and designed the experiments, authored or reviewed drafts of the paper, and approved the final draft.
- Dragi Kocev conceived and designed the experiments, authored or reviewed drafts of the paper, and approved the final draft.
- Sašo Džeroski conceived and designed the experiments, authored or reviewed drafts of the paper, and approved the final draft.

### Data Availability

   The code for constructing the curves is available at GitHub: https://github.com/Petkomat/fr-eval-curves.
   The code for constructing the feature rankings and predictive models used the methods (feature ranking and predictive modeling) implemented in Weka 3.6: https://www.cs.waikato.ac.nz/~ml/weka/.
   The following datasets are available from the UCI repository: https://archive.ics.uci.edu/ml/datasets.php
   Credit approval, Arrhythmia, Balance Scale, Breast Cancer Wisconsin (Original), Breast Cancer, Car Evaluation, Chess (King-Rook vs. King-Pawn), Statlog (German Credit

Data), Statlog (Heart), Heart Disease (for Cleveland and Hungarian data), Hepatitis, Image Segmentation, Ionosphere, Iris, Connectionist Bench (Sonar, Mines vs. Rocks), Tic-Tac-Toe Endgame, Congressional Voting Records, Waveform Database Generator (Version 1), Wine.

The Pima Indians Diabetes data (previously available at the UCI repository) is now available at OpenML: https://www.openml.org/d/37.

The following datasets are available from the Bioinformatics Laboratory: https://file.biolab.si/biolab/supp/bi-cancer/projections/index.html.

AML prognosis, bladder cancer, breast cancer, childhood ALL, CML treatment, breast & colon (colon part of the data set), DLBCL, leukemia, MLL, prostate, SRBCT.

Three additional datasets (aapc, diversity, water) are available at GitLab: http://source.ijs.si/data/classification-data.

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
