# Peer review of "Error curves for evaluating the quality of feature rankings"

_PeerJ Computer Science, doi:10.7717/peerj-cs.310_

## Round 0.1 · original submission · Minor Revisions

The paper is almost ready to be accepted. Please prepare a new version addressing the issues pointed by the reviewers.

Reviewer 1 ·

Basic reporting

No comment

Experimental design

No comment

Validity of the findings

No comment

Additional comments

The paper is scientifically sound and well written, with an exhaustive set of experiments on a suite of different synthetic and real datasets. I have only two points I would like to see better clarified
- there is a non-negligible overlap with Slavkov, 2018; although this is acknowledged by the authors, I think they should stress more the novelty of the present contribution w.r.t. their preliminary work;
- the number of alternatives shown (e.g. the different predictive models for building the FFA/RFA curves) provides a broad and solid landscape, but they are not helping the researcher wanting to apply the proposed method (e.g. what should I do when different models provide very different rankings? Which model should I trust). I think that the paper would greatly benefit by adding a working recipe schema for ta practitioner wanting to practically use the method in an effective way.

Finally, the main text is quite dense and rich in figures - I would rather move more material in the Appendix/SuppMat to better highlight the key messages in the main text avoiding them to sink in a sea of results which may prevent a non-specialistic reader to fully grasp the overall meaning.

Reviewer 2 ·

Basic reporting

The authors address an interesting problem within the context of feature ranking. They present a method for evaluating feature ranking algorithms. The method is based on computing the correctness of the feature ranking based on the accuracy achieved with it using different learning algorithms. Based in two different chains and a simple idea it seems to provide meaningful information about the rankings.

The state-of-the-art work is very complete, although I would suggest to include more recent references about approaches to measure feature stability.


The contribution is stated clearly.

Experimental design

Extensive experimental work is carried out incluging synthetic and real world dataset.

However, the choice of the best learning methods for building the curves ( SVMs and K-NN) is not well justified. I encourage the authors to clarify this point.

Could this depend on the type of dataset: number of instantes, dimensionality, noise ?

Validity of the findings

Specify the limitations and drawbacks of the proposed method.

Specify if this methods could apply to regression problems.

---

## Round 0.2 · accepted · Accept

Both reviewers consider that the new version for the paper is acceptable. Congratulations

Reviewer 1 ·

Basic reporting

no comment

Experimental design

no comment

Validity of the findings

no comment

Additional comments

All the raised issues have been reasonably solved.

Reviewer 2 ·

Basic reporting

The authors have addressed all the changes that were requested. I recommend the acceptation of the paper.

Experimental design

ok

Validity of the findings

ok